# Cerebellar Purkinje cells control eye movements with a rapid rate code that is invariant to spike irregularity

**Hannah L Payne[1], Ranran L French[2], Christine C Guo[3], TD Barbara Nguyen-Vu[1], Tiina Manninen[1,4], Jennifer L Raymond[1]***

[1]Department of Neurobiology, Stanford University, Stanford, United States; [2]Department of Brain and Cognitive Sciences, University of Rochester, Rochester, United States; [3]Mental Health Program, QIMR Berghofer Medical Research Institute, Queensland, Australia; [4]Faculty of Medicine and Health Technology, Tampere University, Tampere, Finland

**Abstract** The rate and temporal pattern of neural spiking each have the potential to influence computation. In the cerebellum, it has been hypothesized that the irregularity of interspike intervals in Purkinje cells affects their ability to transmit information to downstream neurons. Accordingly, during oculomotor behavior in mice and rhesus monkeys, mean irregularity of Purkinje cell spiking varied with mean eye velocity. However, moment-to-moment variations revealed a tight correlation between eye velocity and spike rate, with no additional information conveyed by spike irregularity. Moreover, when spike rate and irregularity were independently controlled using optogenetic stimulation, the eye movements elicited were well-described by a linear population rate code with 3–5 ms temporal precision. Biophysical and random-walk models identified biologically realistic parameter ranges that determine whether spike irregularity influences responses downstream. The results demonstrate cerebellar control of movements through a remarkably rapid rate code, with no evidence for an additional contribution of spike irregularity.
DOI: https://doi.org/10.7554/eLife.37102.001

*For correspondence:
jenr@stanford.edu

## Introduction

Action potentials, or spikes, are the primary language used by the nervous system. Throughout the brain, the rate at which neurons fire spikes encodes information (*Adrian, 1928*; *Kuffler, 1951*; *Hubel and Wiesel, 1959*; *Wright et al., 1967*; reviewed in *Borst and Theunissen, 1999*), affects the activity of downstream neurons (*Tsodyks and Markram, 1997*; *Bagnall et al., 2008*; *Turecek et al., 2016*), and drives motor output (*Crowell et al., 1968*; *Evarts, 1968*; *Evarts, 1969*; *Henn and Cohen, 1976*; *Georgopoulos et al., 1982*; *Hanes and Schall, 1996*; reviewed in *Ebbesen and Brecht, 2017*), indicating that a rate code is widely used for neural computation. In addition, the precise temporal pattern of spikes can encode information beyond that carried by spike rate, suggesting that a temporal code may also contribute to information processing (*O'Keefe and Recce, 1993*; *Theunissen and Miller, 1995*; *Gawne et al., 1996*; *de Ruyter van Steveninck et al., 1997*; *Stopfer et al., 1997*; *Victor, 1999*; *Stopfer and Laurent, 1999*; *Reich et al., 2000*; *Panzeri et al., 2001*; *Huxter et al., 2003*; *Butts et al., 2007*; *Engineer et al., 2008*; *Gollisch and Meister, 2008*; *Huxter et al., 2008*). However, it is not clear whether and how information encoded in the temporal pattern of spikes is transmitted to downstream circuits to influence behavior. Here we leveraged the experimental advantages of the oculomotor system to analyze whether the temporal pattern of spikes in cerebellar neurons contributes to their control of behavioral output.

Temporal coding can take many forms, which fall into two categories: coding based on the precise timing of spikes within the spike train of an individual neuron, or coding based on the precise timing of spikes in one neuron relative to spikes in other neurons. In the cerebellum, previous work has analyzed both types of spike timing in Purkinje cells, which are the sole output neurons of the cerebellar cortex (*Person and Raman, 2012a*; *Steuber and Jaeger, 2013*; *Sarnaik and Raman, 2018*; reviewed in *De Zeeuw et al., 2011*; *Person and Raman, 2012b*). Here, we investigated the impact of the temporal pattern of spikes within individual Purkinje cells.

It has been hypothesized that the irregularity of Purkinje cell spiking affects the cerebellar control of movements (reviewed in *De Zeeuw et al., 2011*). The irregularity of the interspike intervals (ISIs) is a temporal feature of spike trains that varies along a continuum across brain regions and across individual neurons (*Maimon and Assad, 2009*), from highly regular or clock-like (*Guinan et al., 1972*) to highly irregular or supra-Poisson (*Fernandez and Goldberg, 1971*). Further, in a given neuron, the level of spike irregularity can vary as the neuron processes information (*Barlow and Levick, 1969*; *Young et al., 1988*; *de Ruyter van Steveninck et al., 1997*; *Fairhall et al., 2001*). In Purkinje cells, spontaneous spiking is more regular than a Poisson process (*Häusser and Clark, 1997*; *Shin et al., 2007*), and spike irregularity can vary during the processing of sensory signals (*Shin et al., 2007*). Moreover, pathological alteration of the spike irregularity of Purkinje cells has been proposed as a potential cause of ataxia and other cerebellum-related disorders such as autism. In several different mouse models of ataxia (*Hoebeek et al., 2005*; *Walter et al., 2006*; *Alviña and Khodakhah, 2010b*; *Alviña and Khodakhah, 2010a*; *Stahl and Thumser, 2014*; *Mark et al., 2015*; *Jayabal et al., 2016*) and autism (*Peter et al., 2016*), Purkinje cell spiking is more irregular than in normal Purkinje cells. Moreover, increases in the *regularity* of Purkinje cell spiking, caused by reductions in inhibitory synaptic input (*Wulff et al., 2009*) or maternal exposure to cannabinoids (*Shabani et al., 2011*), have also been associated with motor deficits. Such observations of increased or decreased spike irregularity in Purkinje cells in mouse models of ataxia have inspired the hypothesis that any perturbation of normal spike irregularity may impair the ability of Purkinje cells to reliably transmit information for the control of movement (*Hoebeek et al., 2005*; *Walter et al., 2006*; *Wulff et al., 2009*; *Alviña and Khodakhah, 2010b*; *Alviña and Khodakhah, 2010a*; *Luthman et al., 2011*; *De Zeeuw et al., 2011*). Computer modeling has identified short-term synaptic depression as one potential mechanism that would allow spike irregularity in Purkinje cells to influence their control of postsynaptic targets. Because irregular presynaptic spike trains contain short ISIs that recruit more short-term depression, short-term depression has the potential to reduce the mean synaptic conductance in the postsynaptic target during more irregular spike trains (*Luthman et al., 2011*). However, causal evidence for a direct contribution of irregularity to impaired motor control is mixed. In mouse models of ataxia, treatments that reverse the abnormally high irregularity have reversed motor deficits in some cases (*Alviña and Khodakhah, 2010b*; *Alviña and Khodakhah, 2010a*; *Walter et al., 2006*; *Jayabal et al., 2016*), but not others (*Stahl and Thumser, 2013*). Moreover, the severity of motor deficits in different mouse lines does not always correspond to the severity of the perturbation of Purkinje cell spike irregularity in the relevant region of the cerebellum (*Stahl and Thumser, 2014*).

Studies of pathological alterations in Purkinje cell irregularity in mouse models of ataxia raised the question of whether natural variations in the level of Purkinje cell spike irregularity during normal behavior might impact motor output, in addition to the influence of spike rate. To analyze whether spike irregularity is a component of the neural code used by Purkinje cells to control behavior, we took advantage of the close link between the activity of Purkinje cells in the cerebellar flocculus and motor output. Located just two synapses from the motor neurons that innervate the eye muscles, floccular Purkinje cells are a key node in the sensorimotor transformation of visual and vestibular signals into oculomotor commands. Recording and stimulation studies in a range of species have established an influence of Purkinje cell spike rate on eye movement behavior (*Lisberger and Fuchs, 1978*; *Noda and Suzuki, 1979*; *Stone and Lisberger, 1990*; *Lisberger et al., 1994*; *Lisberger, 1994*; *Kahlon and Lisberger, 1999*; *Nguyen-Vu et al., 2013*; *Katoh et al., 2015*; *Kodama and du Lac, 2016*). The possibility that the spike irregularity of floccular Purkinje cells might also influence eye movements has been suggested by the observation of abnormal oculomotor behavior or oculomotor learning in mouse models of ataxia with abnormal Purkinje cell spike irregularity (*Hoebeek et al., 2005*; *Katoh et al., 2007*; *Katoh et al., 2008*; *Wulff et al., 2009*; *Stahl and Thumser, 2014*). Here, we used a combination of recording, stimulation, and computational

approaches to assess whether and how spike irregularity contributes to the control of eye movements by the cerebellum.

## Results

### Mean spike rate and irregularity both vary with oculomotor behavior

We analyzed spike trains from Purkinje cells recorded in the cerebellar flocculus and ventral paraflocculus of rhesus monkeys and the flocculus of mice while they made smooth eye movements in response to a variety of vestibular and visual stimuli (for full descriptions of the stimuli, see Materials and methods). It is well established that the spike rate of a majority of Purkinje cells in this region of the cerebellum encode gaze velocity (*Lisberger and Fuchs, 1974*; *Lisberger et al., 1994*; *Pastor et al., 1997*; *Raymond and Lisberger, 1998*; *Hirata and Highstein, 2000*; *Katoh et al., 2015*). We assessed whether spike irregularity in these neurons also correlates with gaze velocity, and hence might contribute to the control of gaze.

Consistent with previous work (*Lisberger and Fuchs, 1978*; *Lisberger et al., 1994*; *Pastor et al., 1997*; *Raymond and Lisberger, 1998*; *Hirata and Highstein, 2001*; *Katoh et al., 2015*), the spike rate of Purkinje cells in our samples was highly correlated with gaze velocity. Gaze velocity is defined as the angular velocity of the eye in world coordinates, which is equal to the sum of eye velocity in the head, plus head velocity in the world. When animals tracked a moving visual stimulus with the head stationary, gaze velocity was equal to eye velocity, and there was a clear correlation between Purkinje cell spike rate and gaze velocity, in both the population mean (*Figure 1A left*) and individual cells (*Figure 1B*), in both monkeys (*Figure 1C top*) and mice (*Figure 1C bottom*, *Figure 1—figure supplement 1*). Similarly, when monkeys or mice held their eyes roughly stationary relative to the head to track a visual stimulus that moved with the head, a behavior known as cancellation of the vestibulo-ocular reflex, gaze velocity closely tracked head velocity, and spike rate was again correlated with gaze velocity (*Figure 1A right*, *Figure 1C*, *Figure 1—figure supplement 1*). The same, roughly linear relationship between spike rate and gaze velocity was observed during eye movement responses to multiple combinations of visual and vestibular stimuli with different velocity profiles, and when the visual stimulus used to drive eye movements was a large-field visual stimulus (monkeys and mice) or a small visual target (monkeys only) (*Figure 1C*; mean correlation coefficient across conditions $0.81 \pm 0.02$, $p=10^{-69}$, n = 120 cells, monkey; correlation coefficient $0.71 \pm 0.04$, $p=10^{-19}$, n = 33 cells, mouse).

Like spike rate, spike irregularity was also correlated with gaze velocity across a range of visual and vestibular stimulus conditions in both monkeys and mice (*Figure 1A,B,D*). Spike irregularity was quantified using the coefficient of variation-2 (CV2) of the ISIs, which provides a local measure of the variation in ISIs. In contrast to the coefficient of variation (CV) of the ISIs, the CV2 is not strongly influenced by smooth changes in spike rate. Therefore, CV2 provides a useful measure of irregularity when there are underlying changes in rate (*Holt et al., 1996*; *Shin et al., 2007*; *Wulff et al., 2009*; *Gao et al., 2012*; *Stahl and Thumser, 2014*; *Mark et al., 2015*; *Jayabal et al., 2016*; *Peter et al., 2016*). The CV2 is calculated by taking the absolute difference between each adjacent pair of ISIs, divided by their mean:

$$\mathrm{CV2}_i = \frac{|\mathrm{ISI}_{i+1} - \mathrm{ISI}_i|}{(\mathrm{ISI}_{i+1} + \mathrm{ISI}_i)/2} \qquad (1)$$

where $\mathrm{ISI}_i = t_i - t_{i-1}$, and $t_i$ represents the time of spike $i$. The CV2 can range from 0 (two identical ISIs) to a theoretical maximum of 2 (one small ISI adjacent to one infinitely long ISI), with the mean CV2 of a Poisson process equal to 1 (*Holt et al., 1996*), and higher values of CV2 reflecting more irregular spiking. The CV2 was calculated for each pair of ISIs and interpolated to yield an instantaneous estimate of spike irregularity (see Materials and methods), thus characterizing the irregularity of Purkinje cell spiking on a timescale similar to that for estimating instantaneous spike rate. During visual tracking and VOR cancellation, there was modulation of spike irregularity that mirrored the modulation of gaze velocity, with CV2 increasing for gaze movements in one direction and decreasing for gaze movements in the other direction (*Figure 1A*, *purple*). More generally, across the set of visual and vestibular stimulus conditions tested, a roughly linear relationship between spike irregularity and gaze velocity was observed in individual Purkinje cells (*Figure 1B*, *bottom*; *Figure 1—figure*

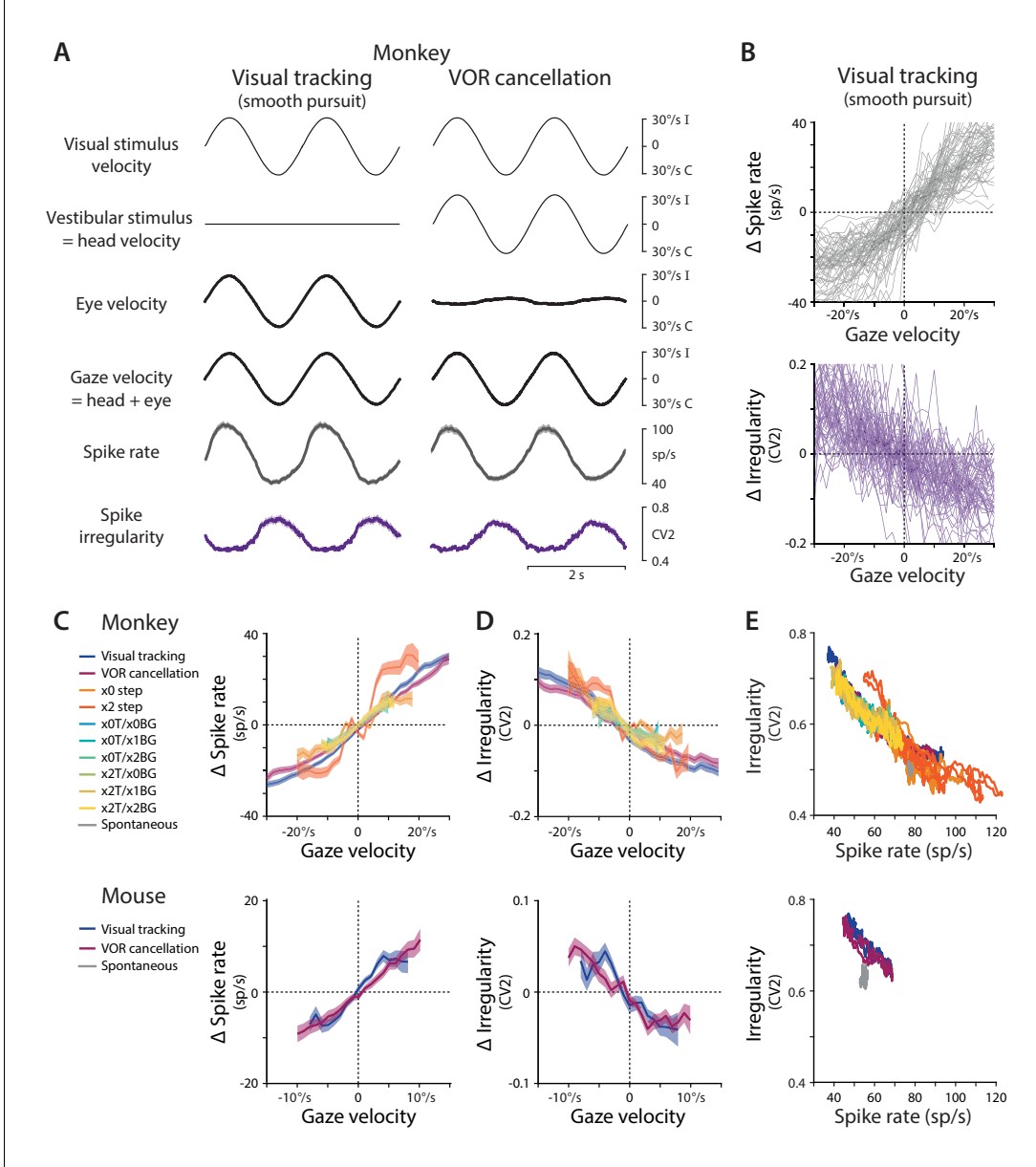

**Figure 1.** Spike rate and irregularity during oculomotor behavior. (A) Visual and vestibular stimuli, oculomotor responses, and Purkinje cell spike rate and local spike irregularity (CV2) during two oculomotor behaviors in rhesus monkeys: visual tracking (smooth pursuit of a small visual target; *left*) and VOR cancellation (cancellation of the vestibulo-ocular reflex, achieved by tracking a visual target that moves exactly with the head; *right*). Upward deflections represent ipsiversive movements of the visual stimulus, head, eye, or gaze; downward deflections represent contraversive movements (relative to the side of the brain in which the Purkinje cell was recorded). Data were averaged across stimulus cycles (median 24 cycles, range 10 to 118) for each neuron, and then averaged across neurons (n = 66 cells from two monkeys, data reanalyzed from *Raymond and Lisberger, 1998*). For clarity of illustration, the average cycle is repeated twice. Results are shown for the largest subclass of Purkinje cells in the cerebellar flocculus, the $E_iH_i$ Purkinje cells, which increase their firing in response to ipsiversive eye motion during visual tracking and ipsiversive head motion during VOR cancellation (*Raymond and Lisberger, 1997*). Results from Purkinje cells with other responses to eye or head motion are shown in *Figure 1—figure supplement 2*. (B) The relationship between spike rate and gaze velocity (*top*) and spike irregularity (CV2) and gaze velocity (*bottom*) during visual tracking for individual Purkinje cells included in the averages in panel (A). Spike rate, CV2, and gaze velocity were first averaged across stimulus cycles, and then binned according to gaze velocity (2°/s bins). The average spike rate or CV2 was subtracted from each cell to obtain Δ spike rate or Δ CV2, respectively. (C) The relationship between spike rate and gaze velocity during eye movement responses to different combinations of vestibular and visual stimuli (see Materials and methods for descriptions of each visual-vestibular stimulus listed in the legend). The Δ spike rate and Δ irregularity were calculated as in (B) and then averaged across the population of all $E_iH_i$ Purkinje cells recorded in monkeys (*top*; n = 120 cells from four monkeys, data reanalyzed from *Raymond and Lisberger, 1998*; *Raymond and Lisberger, 1997*; *Kimpo et al., 2014*) and mice (*bottom*; n = 33 cells from 29 mice, data reanalyzed from *Katoh et al., 2015*). (D) The relationship between spike irregularity (CV2) and gaze velocity averaged across the population of

*Figure 1 continued on next page*

*Figure 1 continued*

Purkinje cells recorded in monkeys (*top*) and mice (*bottom*). (**E**) The relationship between spike rate and spike irregularity (CV2), averaged across the population of Purkinje cells recorded in monkeys (*top*) and mice (*bottom*) and plotted for each 2 ms time point within the stimulus cycle. Spontaneous activity is plotted in gray. In this and all figures, error bars represent ± SEM.

DOI: https://doi.org/10.7554/eLife.37102.002

The following source data and figure supplements are available for figure 1:

**Source data 1.** Electrophysiology data from monkey recorded during oculomotor behavior, including instantaneous firing rate (MATLAB variable name: FR), local irregularity (CV2), and gaze velocity (GAZE).

DOI: https://doi.org/10.7554/eLife.37102.006

**Source data 2.** Electrophysiology data from mouse recorded during oculomotor behavior.

DOI: https://doi.org/10.7554/eLife.37102.007

**Figure supplement 1.** Spike rate and irregularity during oculomotor behavior in mice.

DOI: https://doi.org/10.7554/eLife.37102.003

**Figure supplement 2.** Relationship between spike rate and irregularity for all Purkinje cells.

DOI: https://doi.org/10.7554/eLife.37102.004

**Figure supplement 3.** Contribution of a refractory period to the relationship between instantaneous spike rate and irregularity.

DOI: https://doi.org/10.7554/eLife.37102.005

*supplement 1*) and in the population means (*Figure 1D*) recorded in monkeys and mice (mean correlation coefficient across conditions –0.53 ± 0.03, p = $10^{-38}$, n = 120 cells, monkey; –0.47 ± 0.05, p = $10^{-10}$, n = 33 cells, mouse).

The observation that the mean spike irregularity of floccular Purkinje cells, like the mean spike rate, was correlated with oculomotor behavior raised the possibility that both features of the spike train might contribute to the control of eye movement behavior in wild type mice. However, the respective contributions of spike rate and irregularity could not be dissociated based on the responses quantified by averaging across trials, since mean spike rate and mean irregularity covaried. Whenever mean spike rate was high, mean CV2 was lower (spike timing was more regular), and when mean spike rate was low, mean CV2 was high (spike timing was more irregular) (*Figure 1E*). The inverse relationship between spike rate and CV2 was observed in the Purkinje cells that encode ipsiversive gaze velocity, which make up the largest subset of Purkinje cells in the cerebellar flocculus (plotted in *Figure 1*), and also in Purkinje cells with different responses to eye velocity or vestibular input (*Figure 1—figure supplement 2*). This covariation between spike rate and irregularity may reflect, in part, a tendency for the refractory period to cause spiking to be more regular at high rates (*Holt et al., 1996*); however, simulations of Poisson spiking with an imposed refractory period did not fully capture the correlation between spike rate and irregularity (*Figure 1—figure supplement 3*; see *Holt et al., 1996*), and several additional factors are thought to influence spike irregularity in Purkinje cells, including voltage-gated conductances (*Raman and Bean, 1997*; *Raman and Bean, 1999*; *Hoebeek et al., 2005*; *Alviña and Khodakhah, 2010b*; *Alviña and Khodakhah, 2010a*; *Gao et al., 2012*; *Stahl and Thumser, 2014*) and synaptic input (*Häusser and Clark, 1997*; *Jaeger and Bower, 1999*; *Shin et al., 2007*; *Wulff et al., 2009*; *Jelitai et al., 2016*; *Peter et al., 2016*). Indeed, over short timescales within individual trials, spike rate and CV2 did not covary strongly (*Figure 2—figure supplement 1*), and we leveraged this dissociation to assess the contribution of each of these features of the spike trains to the control of eye movements.

## Spike rate, but not irregularity, predicts moment-to-moment variations in eye velocity

Moment-to-moment variations in neural activity and behavior allowed the contribution of spike irregularity to be distinguished from that of spike rate (*Figure 2*, *Figure 2—figure supplement 1*). For each Purkinje cell, residual spike rate, residual CV2, and residual eye velocity were calculated by subtracting the corresponding trial mean from the response on each individual trial, computed in 50 ms time bins (*Figure 2A,B*). The ability of spike rate and spike irregularity (CV2) to predict eye velocity were then statistically compared using a linear mixed effects model (see Materials and methods). This analysis found a significant contribution of residual spike rate, but not residual CV2, for predicting residual eye velocity in monkeys (rate: p=0.006; CV2: p=0.56; interaction: p=0.17; likelihood ratio test) and mice (rate: p=0.035; CV2: p=0.38; interaction: p=0.46) (*Supplementary file 1*; see

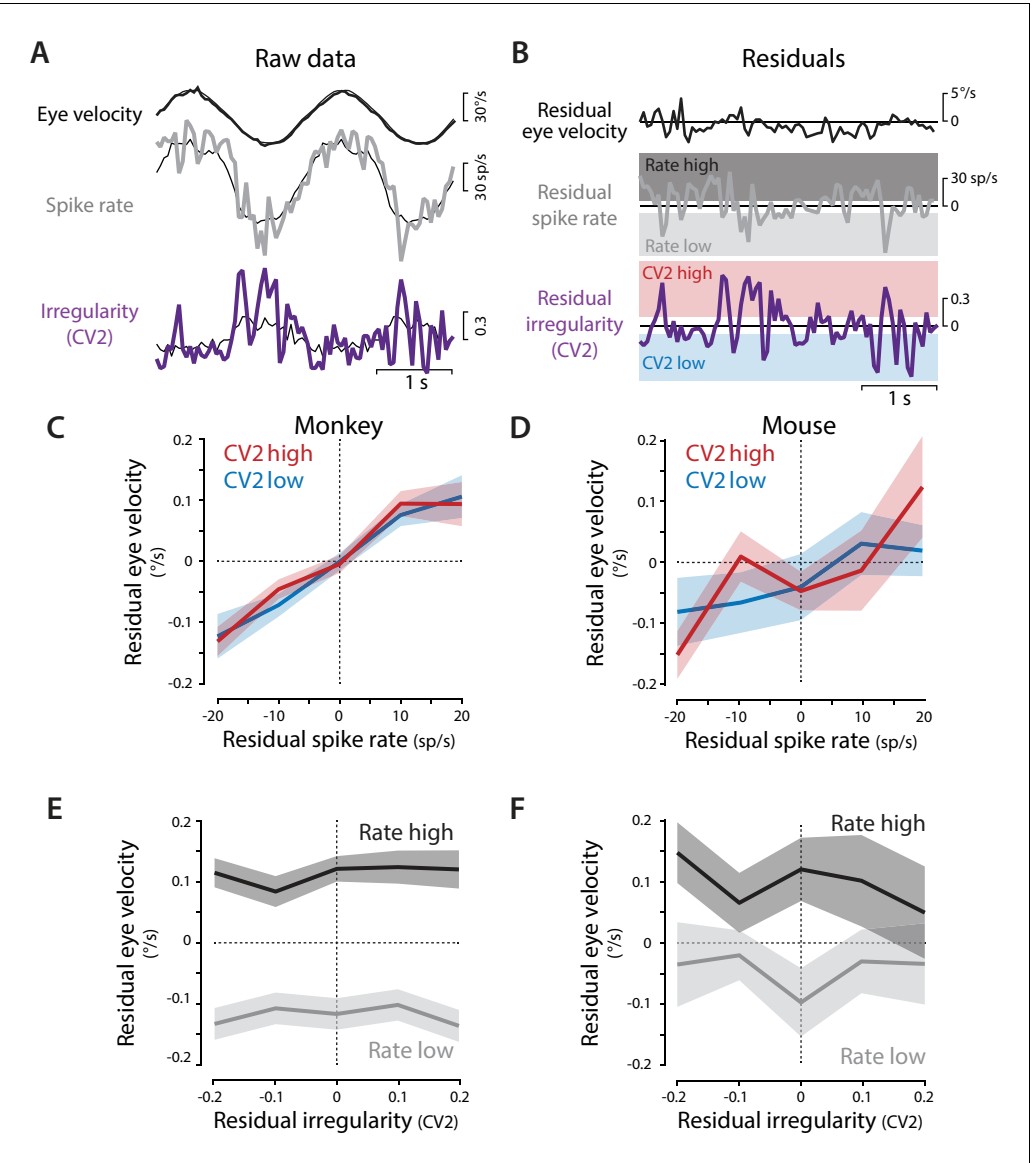

**Figure 2.** Residual analysis of spike rate and irregularity during oculomotor behavior. (A) Example traces of raw eye velocity, instantaneous spike rate, and spike irregularity (*thick lines*) during two cycles of sinusoidal visual tracking in monkey, calculated in 50 ms bins, and overlaid on the mean responses across all cycles for this Purkinje cell (*thin black lines*). (B) Residuals were calculated by subtracting the mean response from the raw response. For visualization only (panels C–F) the residual spike rates and CV2 values were divided into the lower and upper thirds of the distribution for each cell (*shading*). Statistical analysis was conducted on the full distribution of residuals (see text). (C,D) The relationship between residual spike rate and residual eye velocity, plotted separately for time bins with residual CV2 values in the lower (*blue*) or upper (*red*) third of the CV2 distribution for each cell, and then averaged across Purkinje cells for monkeys (C, n = 120 cells from four monkeys) and mice (D, n = 33 cells from 29 mice). (E,F) The relationship between residual spike irregularity (CV2) and residual eye velocity, plotted separately for time bins with residual spike rate values in either the lower (*light gray*) or upper (*dark gray*) third of the spike rate distribution for each cell, for monkeys (E) and mice (F).

DOI: https://doi.org/10.7554/eLife.37102.008

The following figure supplement is available for figure 2:

**Figure supplement 1.** Moment-to-moment spike rate and spike irregularity are only weakly correlated.
DOI: https://doi.org/10.7554/eLife.37102.009

Materials and methods). Spike rate was therefore a better predictor of moment-to-moment eye velocity than spike irregularity.

To visualize this result, the relationship between residual eye velocity and residual spike rate was plotted separately for time bins in which the CV2 fell into the lower or upper third of the CV2 distribution for each cell (*Figure 2B bottom, red vs. blue shading*; *Figure 2C,D*, *red vs. blue traces*). There was a consistent correlation between residual spike rate and residual eye velocity (*Figure 2C, D*): on trials with higher than average spike rate at a particular time within the trial, eye velocity was also higher than average at that time within that trial. One extra spike in a 50 ms time bin, corresponding to a 20 sp/s fluctuation in spike rate, was associated with a roughly 0.1°/s change in eye velocity, consistent with the correlation between single spikes and visual tracking behavior reported by *Chaisanguanthum et al. (2014)*. The same relationship between residual spike rate and eye velocity was observed for time bins with high or low CV2 (*Figure 2C,D*), consistent with the conclusion of the linear mixed effects model that spike irregularity did not substantially modulate the effects of spike rate on eye movements (no main effect of CV2 or interaction between spike rate and CV2, *Supplementary file 1*). Moreover, when residual eye velocity was plotted against residual CV2, there was no correlation (*Figure 2E,F*). Therefore, although on average, both spike rate and irregularity were correlated with each other and with motor output during eye movement behaviors (*Figure 1*), moment-to-moment variations in eye velocity could be explained by moment-to-moment variations in spike rate, but not spike irregularity. Thus, the recording results found no evidence that spike irregularity influenced motor output during oculomotor behavior.

## Optogenetically-driven Purkinje cell spike rate controls motor output independent of spike irregularity

Stimulation experiments provided a causal test of the relationship between spike irregularity and oculomotor behavior. Optogenetic stimulation of Purkinje cells made it possible to independently manipulate spike rate and irregularity by delivering sequences of light pulses with different stimulus rates and irregularities. Channelrhodopsin-2 (ChR2) was selectively expressed in Purkinje cells by crossing a conditional ChR2 mouse line (*Madisen et al., 2012*) with the L7/Pcp2-Cre Jdhu line, which is highly selective for Purkinje cells (*Zhang et al., 2004*; *Witter et al., 2016*). Purkinje cells expressing ChR2 were stimulated in vivo using 500 ms trains of 1 ms pulses of light delivered at mean rates from 20 Hz to 100 Hz. For a given mean rate, the exact same number of pulses were delivered during each train, with variations only in the irregularity of the stimulus pulse timing, so that the CV of each stimulus train was either 0 (perfectly regular), 0.5 (intermediate irregular), or 1 (approximately Poisson irregular) (see Materials and methods). In these experiments, the more common CV measure was used to characterize the irregularity of the stimulus trains rather than CV2, since there was no additional variation in rate. For the irregular stimulus trains, 60 different patterns were randomly generated for each frequency ('*non-repeated*'). In addition to the non-repeated trains, for a subset of stimulus frequencies (20 Hz, 60 Hz, and 100 Hz) one representative irregular (CV = 1) stimulus pattern was repeated ('*repeated*') to provide a measure of trial-to-trial variability, and to allow visualization of the average response to an irregular train. Because Purkinje cells have high rates of spontaneous spiking, the actual spike irregularity and rate achieved in Purkinje cells differed from that of the stimulus trains, and thus was measured explicitly (see below). Stimulus trains were delivered unilaterally to the flocculus of awake, head-restrained mice in a completely dark room, and the resulting eye movements were measured.

Extracellular recordings confirmed that optogenetic stimulation effectively dissociated spike rate and irregularity in Purkinje cells in vivo. Purkinje cells in the flocculus were identified based on their reliable, short-latency responses to 1 ms light pulses (median first-spike latency 1.24 ± 0.07 ms, standard deviation 1.36 ± 0.17 ms, n = 13, 1 mW light intensity; *Figure 3A–C*), and their identity was confirmed by their characteristic spike waveform (*Figure 3A*), high-frequency spontaneous activity (62.2 ± 6.9 sp/s, n = 13), and recording location within the layers of the cerebellar cortex. Putative interneurons were also recorded, which did not have short-latency responses to light pulses (*Figure 3B,C*), and which had different spike waveforms, spontaneous rates, and/or recording locations than the Purkinje cells. Since the interneurons can influence behavior only via their effects on the output of Purkinje cells, interpretation of the relationship between Purkinje cell spiking and motor output is not affected by the activity of the interneurons, hence they were not analyzed further.

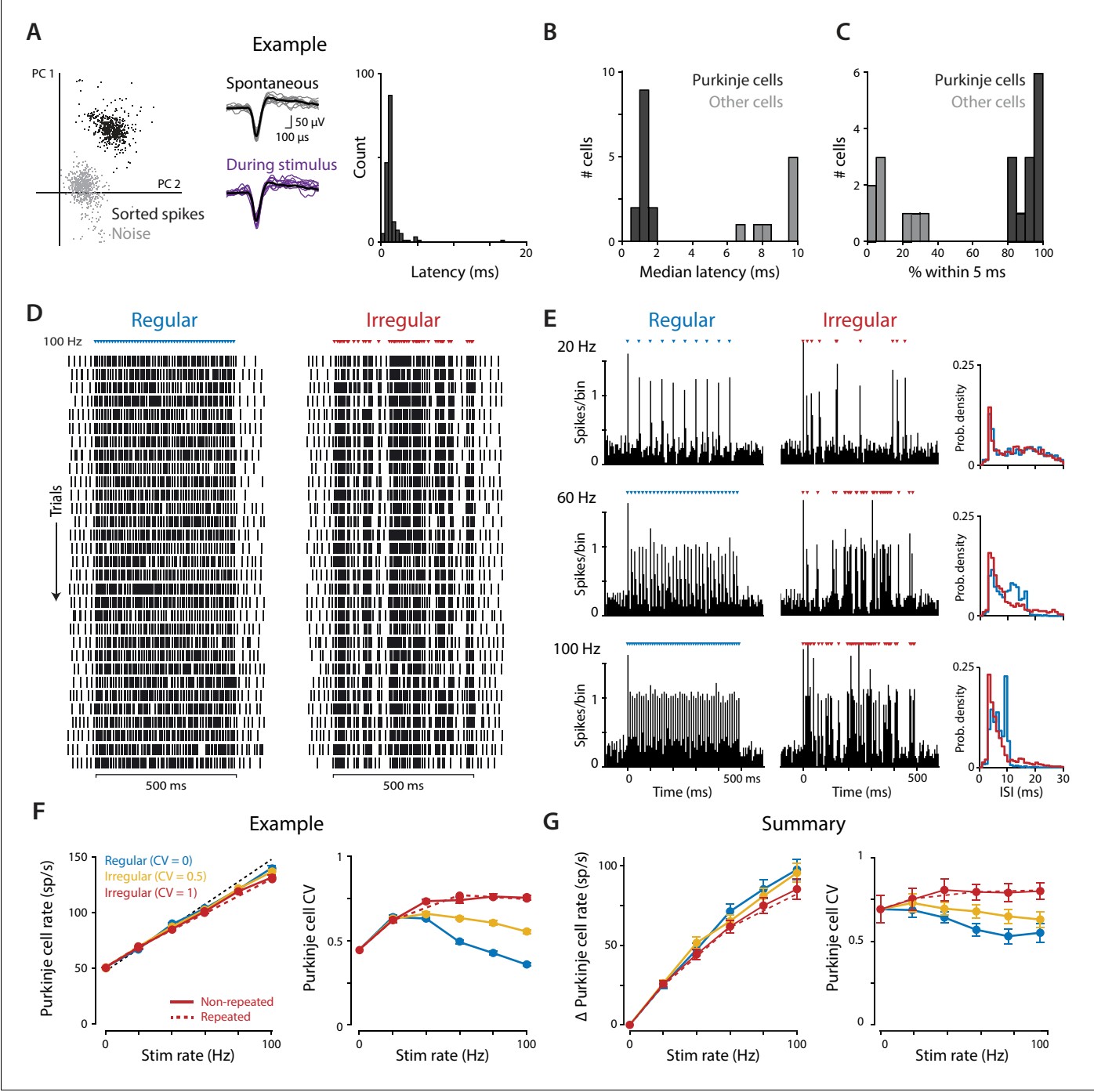

**Figure 3.** Optogenetic stimulation of Purkinje cells with regular and irregular stimulus trains. (**A**) Recordings of optogenetically-driven spikes in an example Purkinje cell. *Left*, first two principle components of waveforms sorted as noise (*gray*) or spikes (*black*). *Middle*, spontaneous and optogenetic stimulus-driven spike waveforms. *Right*, distribution of the latency to the first spike following a single light pulse (1 ms, 1 mW). (**B**) Histogram of median first-spike latencies following a single light pulse for the population of identified Purkinje cells and for other cerebellar cell types (1 ms, 1 mW light pulses). (**C**) Histogram of the percent of trials for which the first spike following the first light stimulus occurred within 5 ms, for Purkinje cells and other cell types. (**D**) Raster of spike times in the example Purkinje cell shown in (**A**) during 100 Hz trains of regular (CV = 0; *left*) and irregular (CV = 1, *repeated; right*) optogenetic stimulation. Colored triangles above rasters indicate the time of each 1 ms light pulse. Each row represents a single trial, with 31 trials shown for each stimulus train, spanning 60 min of recording. (**E**) Peristimulus time histograms (*left*; 5 ms bin width) and ISI distributions (*right*; 1 ms bin width) for the example cell during 20 Hz, 60 Hz, and 100 Hz regular and irregular optogenetic stimulation. (**F**) Mean spike rate (*left*) and CV (*right*) for the example cell as a function of stimulus rate. The black dashed line represents a 1:1 relationship between stimulus rate and the increase in Purkinje cell spike rate relative to the spontaneous baseline. Solid lines show the responses to the regular and irregular (*non-repeated*) stimuli. The

*Figure 3 continued on next page*

*Figure 3 continued*

red dashed line shows the response to the irregular (CV = 1, *repeated*) trains shown in (D,E). (G) Mean increase in spike rate relative to baseline (*left*) and mean CV (*right*) for the population of Purkinje cells (n = 12, 1 mW light intensity). There was a small but significant effect of stimulus irregularity on Purkinje cell spike rate (main effect of stimulus irregularity $F_{(2,22)} = 20.73$, p=0.0009; interaction between rate and irregularity, $F_{(10, 110)}=5.48$, p<0.0001, two-way repeated measures ANOVA; or $F_{(2,24)} = 4.89$, p=0.017; $F_{(10, 120)}=4.87$, p<0.0001 with inclusion of one additional outlier (not shown, see Materials and methods)). Importantly, there was a significant effect of stimulus irregularity on Purkinje cell spike irregularity (main effect of stimulus irregularity $F_{(2,22)} = 20.73$, p<0.0001; interaction between rate and irregularity, $F_{(10, 110)}=15.07$, p<0.0001, two-way repeated measures ANOVA; or $F_{(2,24)} = 21.0$, p<0.0001; $F_{(10, 120)}=13.6$, p<0.0001 with outlier included).

DOI: https://doi.org/10.7554/eLife.37102.010

The following source data is available for figure 3:

**Source data 1.** Optogenetic stimulus-driven Purkinje cell spike rate and spike irregularity.

DOI: https://doi.org/10.7554/eLife.37102.011

Purkinje cells responded to individual light pulses within each stimulus train (*Figure 3D,E*). The increase in spike rate above baseline, averaged over the 500 ms stimulus period, followed the stimulus rate in a roughly 1:1 manner, even for stimulus rates as high as 100 Hz (*Figure 3F,G*, *left*). Irregular stimulus trains drove slightly lower mean spike rates than regular trains (*Figure 3G*, *left*); this is taken into account in the analysis of the behavioral results below. Importantly, the overall irregularity of the combined spontaneous and optogenetically-driven Purkinje cell spikes consistently reflected the irregularity of the stimulus train (*Figure 3F,G*, *right*), with substantially more irregular spiking during the irregular stimulus trains compared to the regular stimulus trains. Thus, optogenetic stimulation provided differential control of spike rate and spike irregularity in cerebellar Purkinje cells.

Optogenetic stimulation of the Purkinje cells elicited robust eye movement responses. Discrete eye movement responses could be observed in response to each light pulse, even at stimulus frequencies as high as 100 Hz (*Figure 4A,B*). The sequence of light pulses during a train led to a cumulative deviation of the eye from its initial position, with greater total deviations in eye position for higher stimulus rates, and correspondingly higher mean eye velocities. The net eye movement response was quantified and compared across stimulus conditions by calculating the mean eye velocity during the 500 ms stimulus train. Like Purkinje cell spike rate, mean eye velocity increased with stimulus rate (*Figure 4C,D*), consistent with the Purkinje cells controlling eye velocity with a rate code. A similar correlation between stimulus rate and mean eye velocity was observed during both regular and irregular stimulus trains. The mean eye velocity evoked by irregular trains was slightly smaller than that evoked by regular trains; however, the mean Purkinje cell spike rate was also slightly smaller during irregular trains (*Figure 3G*). When mean eye velocity was assessed as a function of the actual mean spike rate recorded in the Purkinje cells rather than the stimulation rate, there was no significant difference in the net efficacy of the regular and irregular spike trains to drive eye movements (*Figure 4E*). Similar results were obtained using regular and irregular stimulus trains at a higher light intensity (10 mW, *Figure 4—figure supplement 1*). Thus, our stimulation results do not support the prediction from previous studies that more irregular Purkinje cell spike trains should have less impact downstream than more regular spike trains (*Luthman et al., 2011*; *Hoebeek et al., 2005*).

## Rapid rate code for eye movements

The net impact of Purkinje cell stimulation, as quantified by averaging eye velocity across the 500 ms period of optogenetic stimulation, did not depend on the irregularity of Purkinje cell spiking. On a finer timescale, the eye movement trajectories clearly depended on the exact temporal pattern of stimulation, with eye movements tracking the occurrence of individual stimuli within the train (*Figure 4A,B*). However, additional analysis demonstrated that these moment-to-moment eye movement trajectories could be well accounted for as a rapid, linear response of the eye to changes in the Purkinje cell population spike rate.

The population spike rate was estimated by combining binned spike histograms across the population of recorded Purkinje cells, aligned on the optogenetic stimulus trains. Optogenetic stimulus pulses drove spikes with similar, precise latencies in all recorded Purkinje cells (*Figure 3B,C*), causing rapid fluctuations in the population spike rate during the optogenetic stimulus trains, which

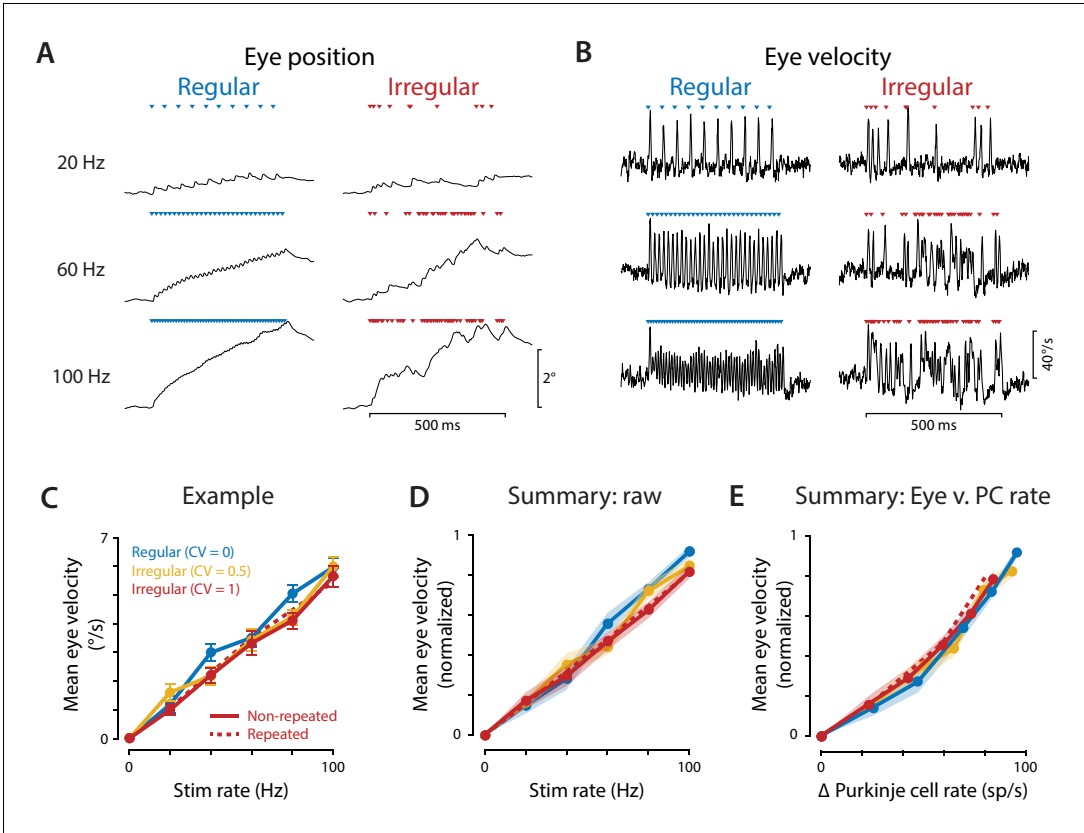

**Figure 4.** Eye movements evoked by regular and irregular Purkinje cell stimulation. (**A**) Eye position averaged across trials in one representative mouse, during optogenetic stimulation of Purkinje cells with regular (*left*) or irregular (CV = 1, *repeated; right*) stimulus trains at 20, 60, and 100 Hz. Colored triangles indicate stimulus times. (**B**) Eye velocity averaged across trials in the example mouse. (**C**) Mean eye velocity, averaged across the 500 ms period of stimulation and plotted as a function of the stimulus rate, for the example mouse. Solid lines represent the results for regular and *non-repeated* irregular trains; dashed line represents the results from the *repeated* irregular trains. (**D**) Mean eye velocity as a function of the optogenetic stimulation rate, averaged across experiments (n = 16 flocculi from 10 mice). Colors indicate the stimulus irregularity, as in (**C**). There was a small but significant interaction of stimulus irregularity and stimulus rate on the mean eye movement responses (fixed effect of irregularity: p=0.34, interaction: p=0.021, likelihood ratio test of linear mixed effects model). (**E**) Mean eye velocity as a function of the mean increase in spike rate recorded in the population of Purkinje cells. Colors indicate the stimulus irregularity, as in (**C**). There was no significant effect of Purkinje cell irregularity on the mean eye movement responses (fixed effect of irregularity: p=0.86, interaction: p=0.80, likelihood ratio test of linear mixed effects model; with the outlier Purkinje cell included in the mean Δ Purkinje cell rate, fixed effect of irregularity: p=0.50; interaction: p=0.88).

DOI: https://doi.org/10.7554/eLife.37102.012

The following source data and figure supplement are available for figure 4:

**Source data 1.** Optogenetic stimulus-driven eye velocity and Purkinje cell spike rates.
DOI: https://doi.org/10.7554/eLife.37102.014

**Figure supplement 1.** Eye movements evoked by regular and irregular Purkinje cell stimulation at a higher light intensity.
DOI: https://doi.org/10.7554/eLife.37102.013

paralleled the rapid fluctuations in eye velocity. We quantified the ability of the population spike rate to predict oculomotor output over these short timescales.

We first determined the *decoding time window* over which the population rate is decoded ('read out') by the downstream oculomotor circuitry. The decoding window is analogous to the encoding window used to analyze rate coding and temporal coding in sensory systems (*Theunissen and Miller, 1995*). The decoding window was evaluated by calculating mutual information to determine

the timescale over which the population spike rate contained information about movements. Mutual information quantified the amount of information (in bits per second) that the Purkinje cell population spike rate conveyed about eye velocity during optogenetic stimulation, as spike rate and eye velocity were smoothed over successively longer time windows (see Materials and methods). Since there is a delay between neural activity and motor output, mutual information was initially calculated for a range of delays, and the delay that yielded the highest mutual information on average, 6 ms, was used in all subsequent analyses. For both regular and irregular stimulus trains, mutual information was maximal when spike rate and eye velocity were smoothed over a temporal window of 3–5 ms (*Figure 5A*). Mutual information decreased when spike rate was smoothed using windows shorter than 3 ms, indicating that fluctuations in spike rate and eye velocity at the shortest timescales were not as strongly related. Mutual information also decreased when the data were smoothed using windows longer than 5 ms, indicating that there is information about eye velocity in the rapid fluctuations of Purkinje cell spike rate that is lost by smoothing over intervals longer than 5 ms. Thus, the Purkinje cell population transmits a rate code that is decoded by the oculomotor circuitry on a timescale of 3–5 ms. Importantly, for rapid timescales encompassing this optimal decoding window of 3–5 ms, the same amount of information about eye movements was transmitted by the Purkinje cell rate code during regular and irregular stimulation (*Figure 5A*). Spike irregularity therefore did not alter the magnitude of the influence of spike rate on eye movements.

Not only did the instantaneous spike rate carry the same amount of information about eye velocity irrespective of spike irregularity, but the transformation between population spike rate and eye velocity could be described by the same linear temporal filter. For each stimulation experiment, a single linear filter was fit using total least squares regression of the eye velocity responses to the *non-repeated* irregular (CV = 1) stimulus trains at all frequencies (20, 40, 60, 80, 100 Hz) against the Purkinje cell responses to the same set of stimuli (*Figure 5B*). The linear filter was then used to predict the eye velocity responses to both the regular and the *repeated* irregular (CV = 1) stimulus trains, which had not been used to fit the filter, by convolving the filter with the Purkinje cell population spike rate during each stimulus train (*Figure 5C*). The linear filter model closely predicted eye velocity (*Figure 5C,D*) with root mean squared error (RMSE) that did not differ significantly between the regular and irregular test stimulus trains (*Figure 5E*). Further, when the model was fit to data from the *regular* stimulation at all frequencies (20, 40, 60, 80, 100 Hz), the resulting linear filter was nearly identical to the filter fit to the irregular data only (*Figure 5—figure supplement 1*). Thus, the same linear filter model described the relationship between spike rate and motor output equally well, regardless of the underlying spike pattern.

The rapid readout of the Purkinje cell rate code could render the circuit sensitive to changes in spike irregularity that alter the precision of the population spike rate (*Walter et al., 2006*). In particular, highly irregular spiking in individual Purkinje cells may degrade the precision of the motor commands carried by the population rate, by increasing its moment-to-moment and trial-to-trial variability. A simple simulation illustrates how spike irregularity in individual neurons can affect the accuracy of the population rate (*Figure 6*). Interspike intervals for a population of 50 simulated Purkinje cells were drawn from Gamma distributions with a given mean rate and level of irregularity. The instantaneous population rate was then calculated in 5 ms temporal windows (population rate = total number of spikes across the population divided by the number of neurons and the window duration). Higher spike irregularity increased the moment-to-moment variation in the population rate from the specified rate, as demonstrated by a broadening of the distribution of instantaneous population rates (*Figure 6A*). This broadening was subtle for variations in spike irregularity in the normal range observed in our recording and stimulation experiments and in previous studies of Purkinje cells in wild type mice (*Figure 6A*, *blue and red*; *Gao et al., 2012*; *Stahl and Thumser, 2014*; *Mark et al., 2015*), but was more pronounced for the higher levels of irregularity that have been reported in mouse models of ataxia (*Figure 6A*, *purple*; *Hoebeek et al., 2005*; *Walter et al., 2006*; *Luthman et al., 2011*; *Gao et al., 2012*; *Stahl and Thumser, 2014*; *Mark et al., 2015*). This variability in the population spike rate could in turn lead to greater behavioral variability, because the behavior tracks rapid fluctuations in the commands carried by the Purkinje cell population rate (*Figure 5A*). The degraded precision of the population rate code is apparent in receiver operating characteristic (ROC) curves representing the discriminability of two motor commands corresponding to target Purkinje cell population spike rates of 60 sp/s and 100 sp/s. The area under the ROC curve decreased with higher spike irregularity in the individual Purkinje cells, indicating

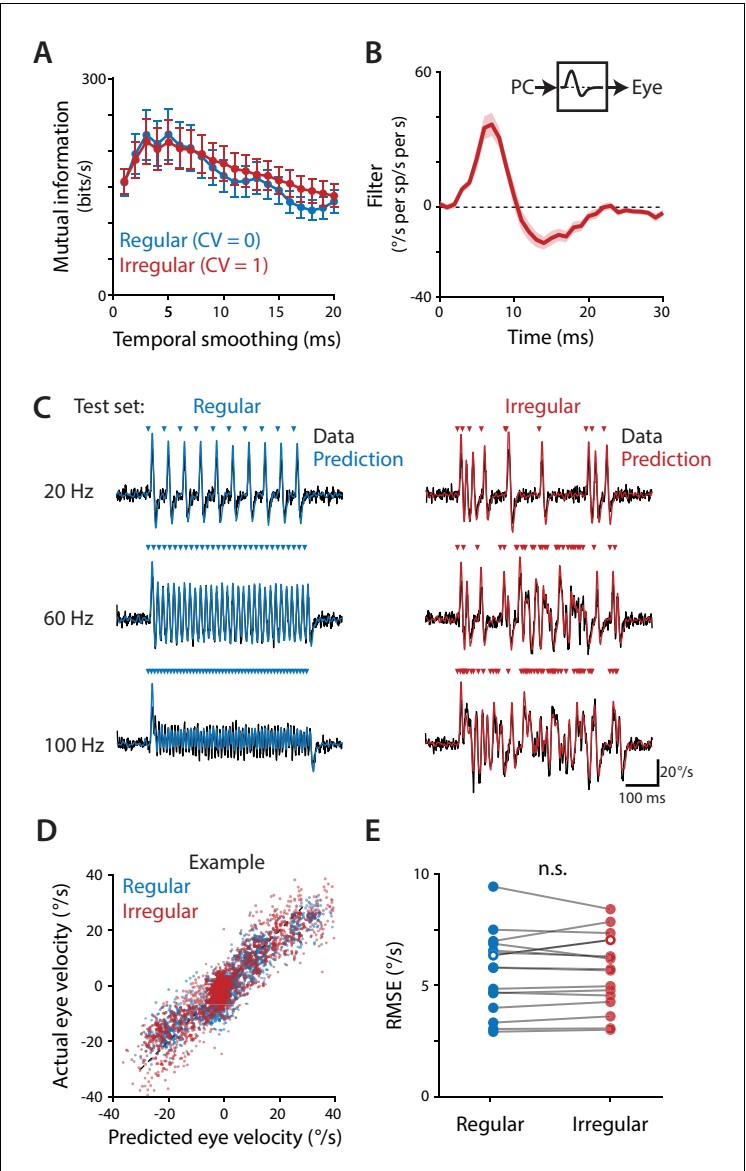

**Figure 5.** Purkinje cells control eye velocity with a rapid rate code. (A) Mutual information between Purkinje cell spike rate and eye velocity during regular and irregular (CV = 1, *repeated*) optogenetic stimulus trains, calculated with the Purkinje cell spike rate smoothed by taking a moving average over time windows of different lengths ('Temporal smoothing'). Purkinje cell spike rate carried the same amount of information about eye velocity during the regular and irregular stimulus trains for temporal smoothing windows from 1 ms through 8 ms (all p>0.1 for 1 ms – 8 ms, 13 ms, and 20 ms; for other window lengths p ranges from 0.0003 to 0.0361, post hoc test with Sidak correction; two-way repeated measures ANOVA with significant interaction between smoothing window length and irregularity (F (19, 285)=11, p<0.0001, n = 16 flocculi from 10 mice; or all p>0.1 for 1 ms – 8 ms, 13 ms, and 20 ms and other p ranging from 0.0002 to 0.0223 with outlier Purkinje cell included). (B) Linear filter describing the transformation of Purkinje cell rate into eye velocity, fit using data from irregular stimulation (CV = 1, *non-repeated;* n = 16 flocculi from 10 mice). (C) Actual eye velocity (*black*) and predicted eye velocity (*colors*) in an example mouse for test stimuli that were not used to fit the linear filter model: regular and irregular (CV = 1, *repeated*) stimulation at 20, 60, and 100 Hz. (D) Actual versus predicted eye velocity at each 1 ms time point during the regular and irregular test stimuli, for the example mouse in (C). (E) Root mean squared error (RMSE) between actual and predicted eye velocity for each experiment (5.57 ± 0.45°/s regular, 5.56 ± 0.41°/s irregular, n = 16, p=0.92, n.s.: not significant, paired t-test; or p=0.87 with outlier Purkinje cell included). RMSE for the example mouse in (C,D) is plotted with hollow circles.

DOI: https://doi.org/10.7554/eLife.37102.015

*Figure 5 continued on next page*

*Figure 5 continued*

The following source data and figure supplement are available for figure 5:

**Source data 1.** Mutual information and linear filter fits for *Figure 5*.

DOI: https://doi.org/10.7554/eLife.37102.017

**Figure supplement 1.** Linear filter models fit to regular or irregular training data.

DOI: https://doi.org/10.7554/eLife.37102.016

decreased ability to discriminate distinct motor commands conveyed by the population spike rate (*Figure 6B*). This provides a potential reconciliation of our current finding of a rapid rate code for motor control with previous reports in ataxic mice suggesting that abnormally high irregularity of Purkinje cell spike trains disrupts normal downstream signaling and motor output (*Hoebeek et al., 2005*; *Walter et al., 2006*; *Wulff et al., 2009*; *Alviña and Khodakhah, 2010b*; *Alviña and Khodakhah, 2010a*; *Luthman et al., 2011*). Pathological levels of spike irregularity may cause ataxia, not through a direct effect of the irregularity itself, but by degrading the accuracy of the rapid rate code carried by Purkinje cells.

## Biophysical model of Purkinje cell synaptic transmission

We used a conductance-based model of the synapses between Purkinje cells and their postsynaptic targets to analyze the cellular properties governing how the temporal properties of Purkinje cell spike trains influence spiking in their target neurons (*Luthman et al., 2011*; *Steuber et al., 2011*). A previous study using this model indicated that more irregular Purkinje cell spike trains should be less effective, on average, than regular spike trains at inhibiting postsynaptic targets (*Luthman et al., 2011*). We analyzed the sensitivity of this finding to the choice of specific model parameters.

The model consisted of a multi-compartmental reconstruction of a Purkinje cell target neuron in the deep cerebellar nucleus (*Luthman et al., 2011*; *Steuber et al., 2011*). The target neuron received inhibition from 50 Purkinje cells (*Person and Raman, 2012a*), each firing spontaneously at 60 sp/s (CV = 0.5), and excitation from 150 mossy fibers, each firing spontaneously at 20 sp/s (CV = 1) (*Figure 7A*), with the inhibitory synapses exhibiting short-term depression as modeled by *Shin et al. (2007)*. The model target neuron also contained Hodgkin-Huxley style ion channel conductances based on experimental measurements, as in the original model (see Materials and methods).

To simulate the optogenetically-driven Purkinje cell activity recorded in vivo, stimulus-driven Purkinje cell spikes following the same regular and irregular stimulus patterns used in the stimulation experiments were superimposed on a background of spontaneous spiking at 60 sp/s (*Figure 7B*). Each stimulus-driven Purkinje cell spike was delayed by a random amount (1.24 ms ± 1.36 ms, mean ± STD for a Gaussian distribution truncated at 0 ms) relative to the common stimulus time to mimic the variability in spike latencies observed in vivo (median: 1.24 ± 0.07 ms, standard deviation: 1.36 ± 0.17 ms, n = 13, *Figure 3B*). The net impact of Purkinje cell stimulation was quantified by computing the spike rate of the model target neuron, averaged over the 500 ms stimulus window, for each stimulus train. The impact of stimulus trains with different rates and irregularity on target neuron output were compared with the empirical effects of the same stimulus trains on eye movements.

When model parameters were identical to those previously published (*Luthman et al., 2011*), the model predicted that irregular stimulus trains (CV = 1) would have substantially smaller net impact on mean target neuron output than regular stimulus trains (CV = 0) (*Figure 7C₁*), as previously reported for simulations of spontaneous Purkinje cell spiking with different levels of irregularity. This prediction was inconsistent with our experimental observation of similar net impact of regular and irregular spike trains on eye movements (*Figure 4*); thus the in vivo observations could not be explained by the previously chosen model parameters. Those parameters had been selected, wherever possible, to match biophysical parameters reported in the experimental literature. However, the values reported for some biophysical parameters differ across experimental studies or cerebellar regions, or are otherwise not well constrained. We found that changes to specific model parameters within a biologically realistic range allowed the model to reproduce the experimental results.

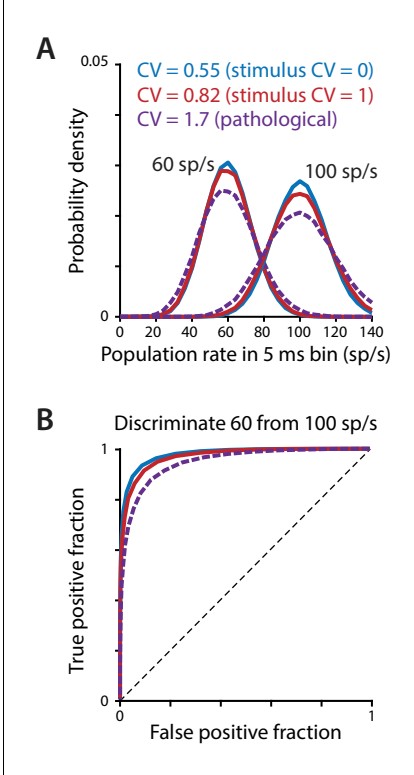

**Figure 6.** Effect of spike irregularity on the variability of the population spike rate. (**A**) The probability density of the population spike rate, computed in 5 ms bins, for a population of 50 simulated Purkinje cells, each firing asynchronously at a given mean rate with the level of irregularity indicated in the legend. The two lower levels of irregularity (CV = 0.55 and CV = 0.82) correspond to the mean CV measured in Purkinje cells during 100 Hz regular or irregular optogenetic stimulation (*Figure 3G*). The high level of irregularity (CV = 1.7) is near the high end of the range reported in mouse models of ataxia (*Hoebeek et al., 2005*; *Gao et al., 2012*; *Stahl and Thumser, 2014*; *Mark et al., 2015*) and was used in the simulations of *Luthman et al. (2011)*. (**B**) Receiver operator characteristic (ROC) curve (*Fawcett, 2006*) for the distributions in (**A**), with the true positive fraction (sensitivity) plotted as a function of the false positive fraction (1 – specificity). Chance performance (no discrimination) is represented by the dashed diagonal line; perfect discrimination is represented by a step function beginning at the origin. Increasing the CV from 0.55 to 0.82 has a minor effect on the ability to discriminate the two frequencies, whereas discrimination is more impaired for a pathological level of irregularity (CV = 1.7).

DOI: https://doi.org/10.7554/eLife.37102.018

One parameter for which different values have been reported in the experimental literature is the amount of short-term plasticity at the synapses from Purkinje cells to their target neurons, which was previously identified as a potential mechanism by which Purkinje cell spike irregularity might influence the activity of the target neurons. Studies in young rodents reported strong (*Pedroarena and Schwarz, 2003*) or moderate (*Telgkamp and Raman, 2002*; *Telgkamp et al., 2004*) short-term depression at these synapses. Such short-term plasticity was previously predicted to cause more irregular spike trains to have reduced net impact on their targets (*Luthman et al., 2011*), contrary to our experimental observations. However, it was recently discovered that short-term plasticity at these synapses is developmentally regulated, such that synaptic transmission in adult mice is remarkably independent of spike rate (*Turecek et al., 2016*; *Turecek et al., 2017*). Our experiments were performed in adult mice; therefore, we eliminated short-term depression in the model. However, elimination of short-term depression was not sufficient to allow the model to reproduce the experimentally observed responses to optogenetic stimulus trains (*Figure 7C$_2$*). Even without short-term depression, the model predicted smaller net impact of irregular than regular stimulus trains, contrary to the experimental observations.

A second mechanism through which the irregularity of Purkinje cell stimulation could influence target neuron spiking is through the creation of synchronous gaps in inhibitory synaptic transmission (*Person and Raman, 2012a*). At high stimulus rates, the regular interstimulus intervals were too short to permit the model postsynaptic neuron to reliably reach threshold and fire a spike before the next synchronous inhibitory input arrived (*Figure 7C*, *top right*). In contrast, highly irregular stimulus trains at the same mean frequency contained a wide range of interstimulus intervals, including some that were sufficiently long to permit postsynaptic firing (*Figure 7C*, *bottom right*). This observation suggested that the irregularity of Purkinje cell stimulation might influence the mean output of the target neurons whenever the strength of Purkinje cell synaptic inhibition was large enough, relative to the strength of excitation, so that one stimulus pulse to the Purkinje cells could suppress spiking in the target neuron for the duration of the mean interstimulus interval. In contrast, floccular Purkinje cell target neurons in the vestibular nuclei are able to sustain high spontaneous firing rates of ~50–120 sp/s despite incessant Purkinje cell inhibition (*Lisberger et al., 1994a*; *Zhang et al., 1995b*; *Zhang et al., 1995a*; *Lisberger et al., 1994b*;

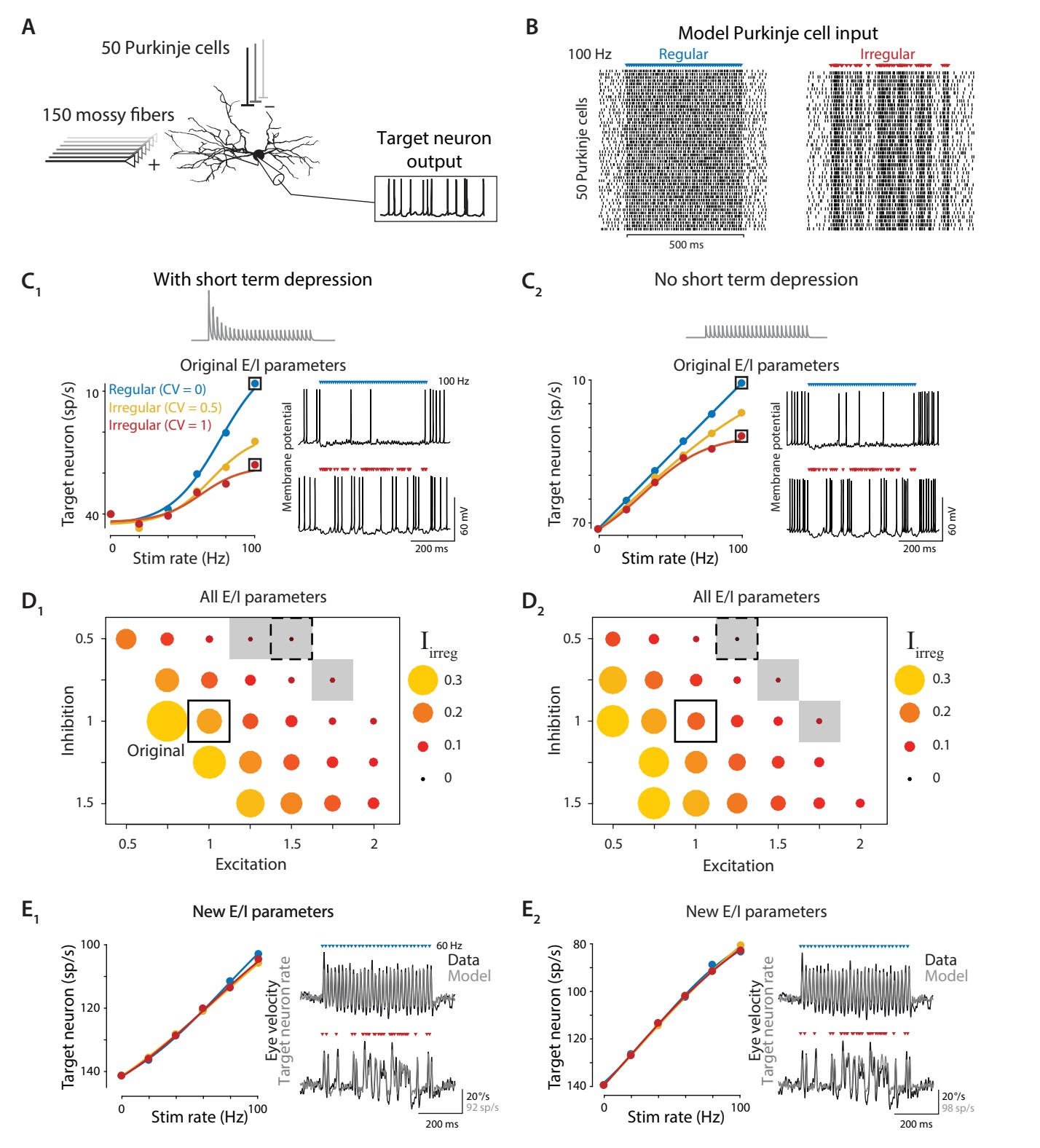

**Figure 7.** Biophysical model: During simulated optogenetic stimulation, impact of spike irregularity is sensitive to the excitation/inhibition ratio. (**A**) Schematic showing elements in the model. A biophysical model of a Purkinje cell target neuron received inhibitory synaptic input from 50 Purkinje cells and excitatory synaptic input from 150 mossy fibers. The effect of Purkinje cell stimulation on the output of the model target neuron was compared with the experimentally observed eye movement responses to Purkinje cell stimulation. (**B**) Spike trains in the 50 model Purkinje cells during simulated optogenetic stimulation (100 Hz regular and irregular stimulation illustrated here) were composed of spontaneous spikes plus stimulus-driven spikes. (**C**)
*Figure 7 continued on next page*

*Figure 7 continued*

Target neuron output in response to simulated optogenetic stimulation, for the model with all original parameters including short-term synaptic depression (STD) at the Purkinje cell-to-target neuron synapses (C$_1$) and for the same model but without STD (C$_2$). Note different scales of vertical axes in C$_1$ and C$_2$. *Gray traces* show STD of the inhibitory conductance in the model during 50 Hz stimulation. Mean target neuron spike rate during the 500 ms stimulus train is plotted for a range of different stimulation rates and levels of irregularity (*left*). In this and subsequent panels, the firing rate of the model target neuron is plotted with the y-axis inverted (lower firing rate plotted higher on the y-axis) since Purkinje activation drives a decrease in target neuron activity below the spontaneous rate (*Lisberger et al., 1994b*), which in turn drives eye movements (*Dufossé et al., 1977*). Voltage traces (*right*) show the membrane potential of the model target neuron during 100 Hz regular (*top*) or irregular (*bottom*) stimulation. (D) Irregularity impact index (I$_{irreg}$, see text) for models with different strengths of excitatory (E) and inhibitory (I) conductance (relative to the original model, indicated by solid box), with short-term depression (D$_1$) or without short-term depression (D$_2$) of the Purkinje cell-to-target neuron synapses. The size and color of the circle reflects the value of I$_{irreg}$ for each set of model E/I parameters, with larger circles representing a larger difference in the impact of regular versus irregular Purkinje cell stimulus trains on the mean target neuron output (positive values reflect a bigger impact of regular trains). Gray shading indicates model parameters for which I$_{irreg}$ < 0.05 (compare with gray shading in *Figure 8A*). Dashed boxes indicate parameters that yielded a close fit to experimental results, shown in (E). (E) Mean target neuron output averaged across the 500 ms stimulus trains (*left*); the moment-to-moment trajectory of the model output (*right, gray*) and actual eye velocity from an example mouse (*right, black*) during 60 Hz stimulus trains, for models with new E/I parameters (indicated by *dashed boxes* in panel D), either with short-term depression (E$_1$) or without short-term depression (E$_2$).

DOI: https://doi.org/10.7554/eLife.37102.019

*Beraneck and Cullen, 2007*; *Bagnall et al., 2008*). Therefore, we systematically varied the excitatory and inhibitory conductances in the model target neuron, while holding other parameters constant, to test how the relative strength of excitation and inhibition (the E/I balance) received by the target neuron influenced the effects of Purkinje cell irregularity.

To summarize the net impact of stimulus irregularity on target neuron output across different model parameters, we introduce a metric, I$_{irreg}$, that captures the normalized difference between the mean target neuron response to regular and irregular stimulation at different frequencies:

$$I_{irreg} = \frac{1}{N_{freqs}} \sum_{i=1}^{N_{freqs}} \frac{r_i^{irreg} - r_i^{reg}}{r^{range}}$$

where $r_i^{reg}$ is the mean target neuron rate in response to regular stimulation (CV = 0) at frequency *i*, $r_i^{irreg}$ is the mean response to irregular stimulation (CV = 1) at the same frequency, and $r^{range}$ is the range of mean target neuron rates observed across the entire set of stimulus trains plus spontaneous activity (max – min). I$_{irreg}$ provides a measure of the effect of irregularity on the overall efficacy of spike trains. Identical mean target neuron responses to regular and irregular stimulation at each frequency would produce an I$_{irreg}$ value of 0, whereas larger positive values of I$_{irreg}$ represent less effective inhibition of target neuron activity by irregular than regular stimulation ($r_i^{irreg} > r_i^{reg}$). Therefore, larger values of I$_{irreg}$ indicate that stimulus irregularity has a greater impact on the control of motor output.

During simulated optogenetic stimulation, the balance of excitatory and inhibitory conductances was a key determinant of whether the net impact of regular and irregular stimulus trains on downstream activity was different, as quantified by I$_{irreg}$. In models with higher E/I ratios, the efficacy of the Purkinje cell spike trains had little dependence on spike irregularity, reflected by small values of I$_{irreg}$ (*Figure 7D*, *small symbols*). Moreover, there were models with small I$_{irreg}$ that closely matched the dynamics of the eye movements observed experimentally in response to the optogenetic stimulus trains (*Figure 7E*). Thus, although previous work with the same biophysical model contended that spike irregularity should affect downstream processing of the Purkinje cells' output (*Luthman et al., 2011*), the biophysics of the Purkinje cell-target neuron synapses are also consistent with a negligible effect of Purkinje cell spike irregularity. Indeed, the original model parameters seem to lie in a small region of E/I parameter space where the effects of irregularity are near maximal (*Figure 7*, *Figure 8*).

The biophysical model was also used to examine how the effects of spike irregularity in individual Purkinje cells might be influenced by spike synchrony across Purkinje cells. Optogenetic stimulation had the advantage of allowing independent experimental manipulation of spike rate and irregularity, but it had the disadvantage of driving spiking that was almost certainly more synchronous across the Purkinje cell population than present during natural sensorimotor signal processing (*Bell and Grimm,*

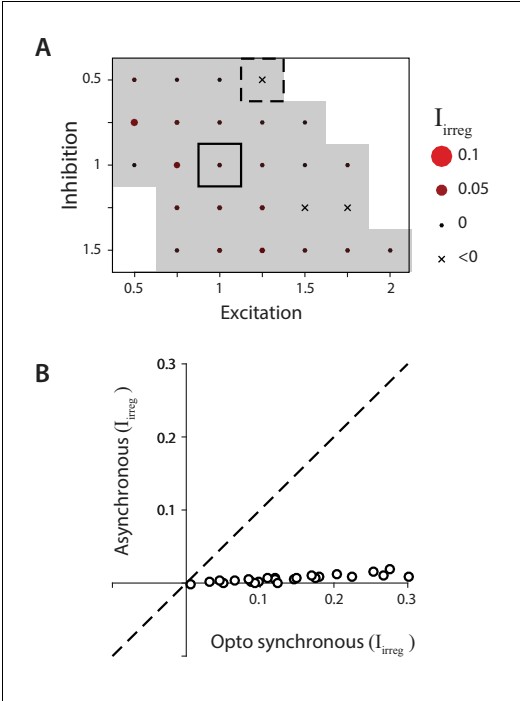

**Figure 8.** Biophysical model: Reduced impact of irregularity during asynchronous activity. (**A**) Impact of irregularity ($I_{irreg}$) during asynchronous Purkinje cell activity for biophysical models without STD. The color scale for $I_{irreg}$ is the same as in **Figure 7**, while the size of the corresponding circles is expanded two-fold to improve visibility of the much smaller differences between models. Gray shading indicates $I_{irreg} < 0.05$, for comparison with **Figure 7D**. (**B**) The impact of irregularity during the more synchronous Purkinje cell activity of simulated optogenetic stimulation ('Opto synchronous'), compared with the impact of irregularity when the same spike trains were shuffled to provide asynchronous Purkinje cell activation in the same model ('Asynchronous'). Note that even in the Opto synchronous condition, there was considerable spike asynchrony due to the independent, spontaneous activity in each Purkinje cell and variation in the latencies of the spikes driven by each simulated light pulse.

DOI: https://doi.org/10.7554/eLife.37102.020

*1969*; *Bell and Kawasaki, 1972*; *Ebner and Bloedel, 1981*; *Shin and De Schutter, 2006*; *Heck et al., 2007*; *de Solages et al., 2008*; *Wise et al., 2010*). The biophysical model was used to predict the impact of Purkinje cell spike irregularity under more natural conditions of less Purkinje cell synchrony than present during optogenetic stimulation. Asynchronous spike trains were generated with the same statistics as the 'synchronous' simulated optogenetic stimulus-driven spike trains by shifting the stimulus-driven spikes in each model Purkinje cell by a random, fixed amount between 0 and 500 ms, with a different random shift for each Purkinje cell in the model (**Figure 8**). For every combination of model parameters, the net impact of irregularity, $I_{irreg}$, was smaller for asynchronous Purkinje cell activity than for the more synchronous activity present during simulated optogenetic stimulation (**Figure 8B**). These smaller values of $I_{irreg}$ for asynchronous rather than synchronous conditions indicate that any effect of spike irregularity on the mean target neuron output under more natural, asynchronous conditions should be *less* than that observed experimentally during more synchronous stimulation. Since the measured impact of Purkinje cell irregularity on mean eye velocity was not significant during synchronous optogenetic stimulation, the model suggests that such irregularity would have a similarly negligible impact under more natural, asynchronous conditions. Taken together, the recording, stimulation, and modeling results make a strong case that floccular Purkinje cells control eye movements with a rapid population rate code, with no evidence for an additional effect of spike irregularity.

# Random walk model of the effect of input variance on target neuron spiking

The biophysical model described above has many parameters, making it impractical to assess how and why each parameter may affect the sensitivity of the model to Purkinje cell spike irregularity. A much simpler, random walk model (*Gerstein and Mandelbrot, 1964*; *Shadlen and Newsome, 1998*; *Salinas and Sejnowski, 2000*) reveals that a range of relationships between spike irregularity and mean target neuron output can arise from even the most basic features of a neuron: a spike threshold, a post-spike reset membrane potential, and a minimum membrane potential, combined with stochastic inputs. In this model, the membrane potential of a neuron is driven by a stochastic input, which represents the sum of all excitatory, inhibitory, and intrinsic currents. The input at each time step is perfectly integrated by the membrane potential until it reaches a threshold, at which point it fires a spike and the membrane potential is reset to a predetermined value (**Figure 9**). A key parameter in the model is the moment-to-moment variability of the stochastic input around its mean value. The irregularity and synchrony of excitatory and inhibitory synaptic inputs are not explicitly modeled, but each can be considered

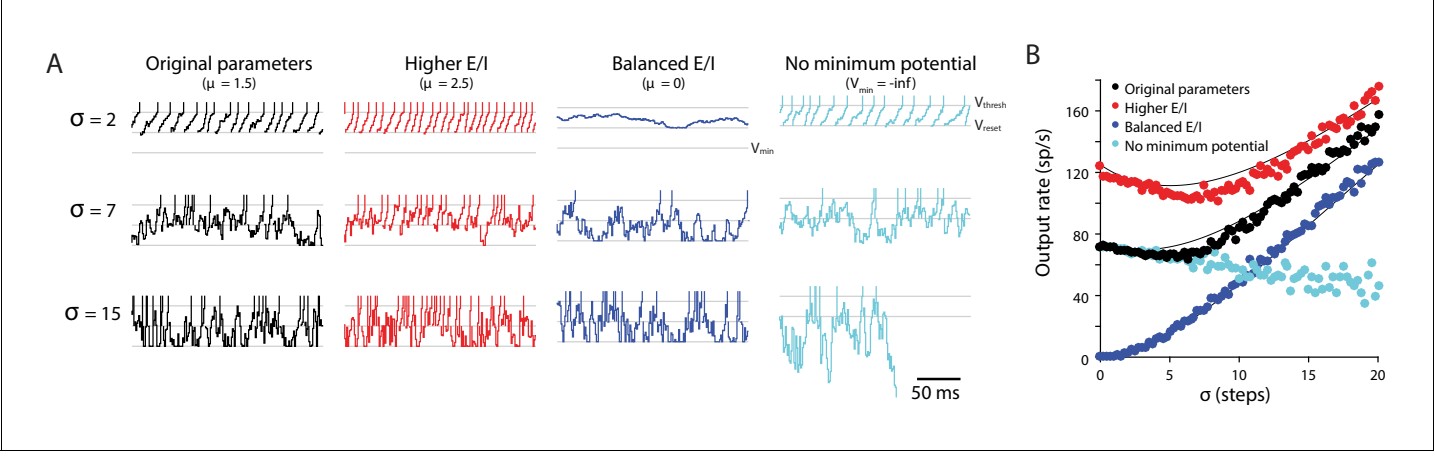

**Figure 9.** Random walk model: Input variability can increase or decrease target neuron firing. Random walk simulation of a simple integrate-and-fire neuron receiving stochastic inputs with mean μ and standard deviation σ. Additional parameters are the threshold voltage $V_{thresh}$ = 40 steps (arbitrary units), reset voltage $V_{reset}$ = 20 steps, minimum voltage $V_{min}$ = 0 and time step dt = 1 ms (**Salinas and Sejnowski, 2000**). *Black:* original parameters (μ = 1.5; $V_{min}$ = 0); *red:* increased E/I ratio ('Higher E/I', μ = 2.5), *dark blue:* balanced excitation and inhibition ('Balanced E/I', μ = 0); *light blue:* membrane potential floor eliminated ('No minimum potential', $V_{min}$ = –infinity, equivalent to raising $V_{thresh}$ and $V_{reset}$ very far from $V_{min}$). (**A**) Example traces showing the membrane potential in model neurons for different levels of input variability (σ = 2, 7, 15 steps). At each time step, the voltage changes by an amount drawn from a normal distribution with mean μ and standard deviation σ. Upon crossing $V_{thresh}$ (*top gray line*), the voltage is automatically lowered to $V_{reset}$ (*middle gray line*) (vertical lines are added to visualize spike times). There is a reflecting floor at $V_{min}$ (*bottom gray line, not present for 'No reflecting barrier'*). (**B**) Numerical simulations (*dots*) and approximate analytic predictions (*lines*) of the firing rate output of model neurons receiving stochastic inputs with mean μ and standard deviation σ. In both the original and 'Higher E/I' simulations, there is a range of σ where small changes in variability have little effect on mean output rates (the range where the curves are approximately flat). Increasing the E/I balance shifts this flat region to the right.

DOI: https://doi.org/10.7554/eLife.37102.021

as increasing the variance of the stochastic input representing the net membrane current (**Figure 6**, **Figure 10**).

Even in the extremely simple random walk model, the effect of increased input variance on the target neuron spike rate depended on the location in parameter space (**Figure 9**). A well-known prior result (**Salinas and Sejnowski, 2000**) is that an increase in input variance can drive a higher spike rate in the target neuron (**Figure 9B**, *positive slopes*). This would correspond, for example, to an increase in Purkinje cell irregularity or synchrony creating more gaps in inhibition that provide a 'window of opportunity' for spiking, thereby making the Purkinje cells less effective at inhibiting spiking in their target neurons. A less recognized finding from the random walk model is that it can also exhibit the opposite behavior, whereby an increase in input variance can decrease spike rate in the target neuron (**Figure 9B**, *negative slopes*; see also **Salinas and Sejnowski, 2000**). Indeed, with other model parameters held constant, as input variance increases from zero, the spike rate can first decrease and then increase (**Figure 9B**, *black and red*). In the transition between these two regimes, changes in input variance within a certain, intermediate range have minimal effect on spike rate in the target neuron (**Figure 9B**, *flat portion of the curves*).

The ability of increased input variance to either increase or decrease firing in the target neuron can be understood by considering two extreme cases. In the case where excitation and inhibition are perfectly balanced (mean input is zero), the target neuron will never reach threshold if there is no input variance. Thus, when excitation and inhibition are balanced, increasing the input variance around the mean of zero increases the target neuron rate (**Figure 9**, *dark blue; monotonically increasing*). A second extreme case is where the mean input is excitatory, but there is no limit to the hyperpolarization of the membrane potential. In the random walk model, this limit is implemented as a 'floor' ($V_{min}$) below which additional inhibitory input has no effect, and which serves to keep the membrane potential relatively close to the threshold. If this floor is removed, then a variable input can cause the membrane potential to wander far from threshold, hence increasing input variance under these conditions decreases the target neuron output on average (**Figure 9**, *light blue;*

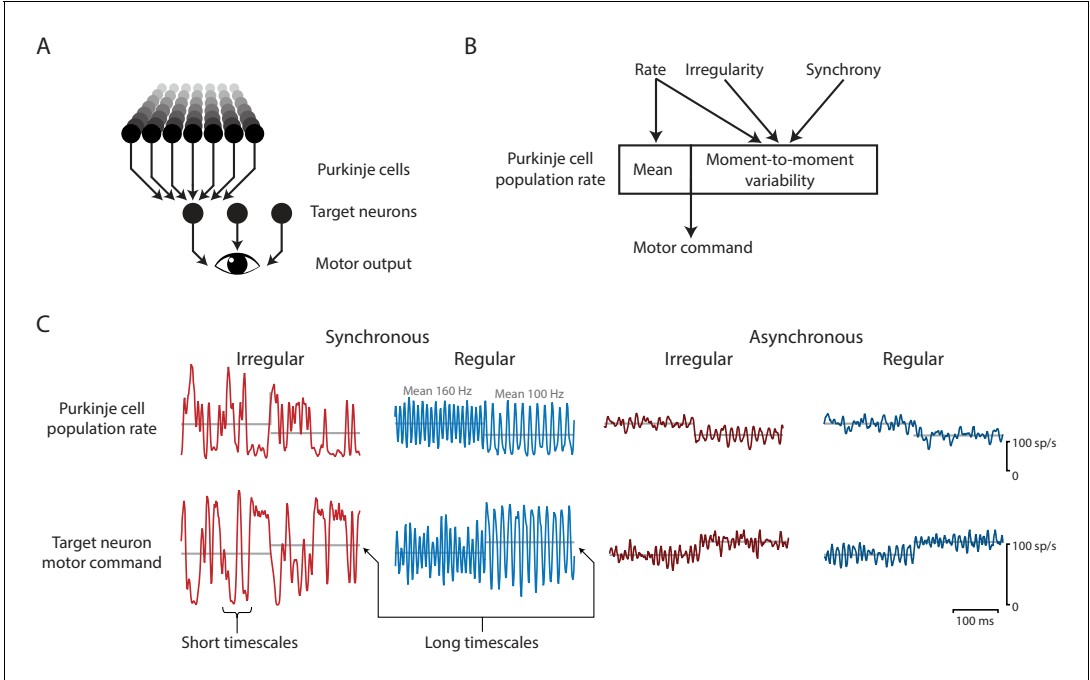

**Figure 10.** Effects of spike irregularity and synchrony on motor output over short and long timescales. Our recording, stimulation, and modeling results indicate that Purkinje cells in the cerebellar flocculus control motor output with a rapid rate code. (**A**) Schematic of the transformation from Purkinje cells to motor output. A population of Purkinje cells (and excitatory neurons, not shown) converges on each target neuron. A population of target neurons drive eye movements. (**B**) The rate, irregularity, and synchrony of spike trains in individual Purkinje cells can each influence the population spike rate. The mean and variability of the Purkinje cell population rate both affect motor output through a rapid rate code. (**C**) Effects of Purkinje cell spike irregularity and synchrony on the Purkinje cell population spike rate (*top*), and target neuron spike rate, representing the eye velocity motor command (*bottom*). The mean population spike rate in a given time interval—either a few milliseconds or hundreds of milliseconds—can predict the mean eye velocity in that same time interval. At short timescales of a few milliseconds, the irregularity and synchrony of spiking in individual Purkinje cells can affect the population spike rate and hence the moment-to-moment motor output. At longer timescales of a few hundred milliseconds, the same mean spike rate elicits the same mean eye velocity (*gray lines*), regardless of the level of irregularity or synchrony.
DOI: https://doi.org/10.7554/eLife.37102.022

monotonically decreasing). Spontaneously active neurons, such as the targets of floccular Purkinje cells, operate in a regime between these two extreme cases, because all neurons have biophysical limits on their membrane potential, and spontaneously active neurons also have net positive mean excitatory drive. In this regime, increasing the input variance can decrease, increase, or have minimal effect on the mean target neuron spike rate, as illustrated by the non-monotonic curves describing target neuron spike rate as a function of input variance (*Figure 9*, *red and black*). Notably, a change in the mean input (e.g. E/I ratio) can alter the sensitivity of spike rate to input variance (*Figure 9B*, *steepness of red vs. black curve*), and shift the absolute level of input variance at which there is a transition from a negative to positive dependence of spike rate on input variance.

This simple simulation demonstrates that biophysical complexity is not necessary to create different relationships between input variance and output spike rate. Instead, the statistics of the random walk itself can yield either an increase or a decrease in output spike rate with an increase in input variance. Our experiments appear to have uncovered a biological example of an overlooked region of parameter space, where changes in synaptic input variability within the range experienced during normal behavior have little impact and hence do not interfere with the rate code for motor output.

## Discussion

The analytical power of the oculomotor system provided an opportunity to test the idea that cerebellar Purkinje cells employ a multiplexed rate code and temporal code to control motor behavior. Previous experimental and modeling studies of ataxia have suggested that pathological changes in

the irregularity of spiking in Purkinje cells influence the responses of downstream neurons and thus behavior (reviewed in *De Zeeuw et al., 2011*). Inspired by these studies, the current research used complementary recording, stimulation, and computational modeling approaches to analyze whether spike irregularity is a component of the neural code used by Purkinje cells to control downstream neurons during normal sensorimotor signal processing and behavior.

The results suggest that the transformation of Purkinje cell activity into motor output is closely approximated by a rapid population rate code (*Figure 10*). Recordings of natural spike trains in Purkinje cells of the cerebellar flocculus during normal oculomotor behavior revealed an intriguing correlation between mean spike irregularity and mean gaze velocity, as also observed for mean spike rate. However, natural, trial-to-trial variations in the eye movement responses to repetitions of the same stimuli could be predicted from fluctuations in spike rate, but not from fluctuations in spike irregularity (*Figure 2*). The ability to predict fluctuations in behavior from the spike rate recorded in a single Purkinje cell suggests that the fluctuations in spike rate are shared across a large fraction of the population of Purkinje cells that control eye movements (see also *Chaisanguanthum et al., 2014*). In contrast, the lack of correlation between fluctuations in CV2 and eye velocity could indicate that 1) the effect of spike irregularity in a single neuron is too small to detect, and fluctuations in spike irregularity are not shared across the population of Purkinje cells, or 2) spike irregularity does not influence eye velocity. The results from the optogenetic stimulation experiments favor the latter interpretation, since the optogenetic stimulus trains drove robust changes in spike irregularity across a large population of stimulated Purkinje cells, yet there was no difference in the mean eye velocity during regular and irregular spike trains after controlling for spike rate. The levels of synchrony across the Purkinje cell population during optogenetic stimulation were almost certainly higher than during natural behavior, however, the biophysical model predicts even smaller effects of spike irregularity under conditions of lower synchrony, consistent with the recordings made under conditions of natural levels of synchrony (*Figure 2*), which also yielded no evidence for an impact of spike irregularity. Taken together, the convergent evidence from the complementary recording, stimulation, and modeling approaches indicates that the irregularity of spiking in Purkinje cells is not a significant component of the code used by the cerebellar flocculus to control normal eye movements. The recording and stimulation results can be fully accounted for by the Purkinje cells controlling eye movements primarily or exclusively via a rate code.

## A remarkably rapid rate code for motor control

### Linear rate coding

Previous studies have demonstrated that electrical or optogenetic manipulation of Purkinje cell activity can drive motor responses, with bigger behavioral responses for higher intensity stimulation (*Ron and Robinson, 1973*; *Belknap and Noda, 1987*; *Lisberger, 1994*; *Wada et al., 2014*; *Nguyen-Vu et al., 2013*; *Heiney et al., 2014*; *Lee et al., 2015*; *Stahl et al., 2015*; *Sarnaik and Raman, 2018*). We extended this work by showing a roughly linear relationship between Purkinje cell spike rate and behavior for optogenetic stimulation-driven spike rates from 0 to 100 sp/s above the spontaneous rate (*Figure 4E*), consistent with the linear coding described in recordings of neural activity during oculomotor behavior (*Figure 1C*; *Shidara and Kawano, 1993*; *Lisberger, 1994*; *Kahlon and Lisberger, 1999*; *Medina and Lisberger, 2007*; *Medina and Lisberger, 2009*; *Katoh et al., 2015*; *Hong et al., 2016*; *Herzfeld et al., 2015*; *Sun et al., 2017*; reviewed in *Raymond and Medina, 2018*).

### Rapid rate coding

The speed of the rate code conveyed by Purkinje cells for controlling eye movements is remarkable. Previous work has shown that inhibitory synaptic transmission between Purkinje cells and their postsynaptic targets is fast, with a synaptic time constant of 2 ms (*Najac and Raman, 2015*). We found that this speed is largely maintained in the downstream oculomotor circuitry, which includes signaling from the flocculus target neurons in the vestibular nuclei to their interneuron and motor neuron targets in the abducens nucleus, and the effects of the latter on the oculomotor plant (reviewed in *Highstein et al., 2004*). Discrete eye movement responses could be observed in response to each 1 ms optogenetic stimulus pulse, for stimulus frequencies at least as high as 100 Hz (*Figure 4*). The mutual information between Purkinje cell rate and eye velocity peaked for a temporal smoothing

window of 3–5 ms (*Figure 5A*). Likewise, the linear temporal filter describing the transformation of Purkinje cell rate into eye velocity had a width of ~5 ms (width at half-amplitude of the initial transient; *Figure 5B*). Thus, the population spike rate in the floccular Purkinje cells appears to be read out by the downstream oculomotor circuitry on a timescale of 3–5 ms. This is less than the typical ISI in an individual Purkinje cell, but is a reasonable timescale for considering the population spike rate in the ~50 Purkinje cells converging on a given target, which will collectively emit 10 or more spikes, on average, per 5 ms interval; likewise, 3–5 ms is a reasonable timescale for considering spike rate in the full population of Purkinje cells controlling eye movements. Such rapid rate coding is likely not unique to the oculomotor system; there is evidence that other motor circuits can transmit information on a similar timescale (*Tang et al., 2014*; *Srivastava et al., 2017*; *Brown and Raman, 2018*).

## Rapid rate coding can endow sensitivity to spike irregularity and synchrony

The rapid speed at which the population spike rate is read out could endow the circuit with sensitivity to any spiking statistics that alter the population rate over short timescales, including spiking statistics that are typically associated with temporal coding, such as spike irregularity and synchrony (*Figure 10*). The convergence of ~50 Purkinje cells onto each target neuron can average out the 'noisiness' of signals carried by the spike trains of individual neurons. Our results indicate that this averaging out needs to occur on a timescale of 3–5 ms, because variations in the population rate get faithfully transmitted all the way to the behavior with this temporal resolution (*Figure 5*, *Figure 10C*). This provides a mechanism by which the high level of Purkinje cell spike irregularity in ataxia (CV of 0.7–2.3 in ataxic mice compared to 0.4–0.6 in control mice; *Hoebeek et al., 2005*; *Gao et al., 2012*; *Stahl and Thumser, 2014*; *Mark et al., 2015*) could degrade the accuracy of movements: by degrading the accuracy and precision of the motor commands carried by the population spike rate (*Figure 6*). In contrast, smaller variations in spike irregularity within the normal physiological range may be well tolerated because they have minimal impact on the population spike rate. Similarly, Purkinje cell synchrony (*Bell and Grimm, 1969*; *Bell and Kawasaki, 1972*; *Ebner and Bloedel, 1981*; *Shin and De Schutter, 2006*; *Heck et al., 2007*; *de Solages et al., 2008*; *Wise et al., 2010*; *Person and Raman, 2012a*; *Sarnaik and Raman, 2018*; *Brown and Raman, 2018*; *Tang et al., 2019*) can be reframed as rapid changes in the Purkinje cell population firing rate on a timescale of a few milliseconds, which could be transmitted downstream via the rapid population rate code for eye velocity. Thus, rapid fluctuations in the population rate, which can be generated by spike irregularity or synchrony, can be read out as rapid fluctuations in the intended motor command (*Figure 10*), allowing spike irregularity and synchrony to influence behavior via the rate code.

## No evidence for additional influence of spike irregularity

Extremely rapid rate coding might seem to blur the line between rate coding and temporal coding, but it is important to distinguish rapid rate coding from true temporal coding. Although the term 'temporal code' is sometimes used to describe changes in spike rate co-occurring with changes in a dynamic stimulus or behavior (e.g. *Gooler and Feng, 1992*), we would reserve the term 'temporal coding' to refer to an influence of the temporal statistics of spiking that *cannot be accounted for by the population spike rate* computed over a biologically relevant timescale (*Theunissen and Miller, 1995*). We found no evidence for temporal coding in the latter, more rigorous sense of the term. Temporal coding could potentially take many forms, and we evaluated just one form based on spike irregularity, hence our results do not rule out other forms of temporal coding in the cerebellar circuit. Our experiments were designed specifically to detect effects of spike irregularity above and beyond what could be accounted for by a population rate code for eye velocity, and we found no evidence for such effects.

Previous work had suggested that more irregular spiking in Purkinje cells would be less effective in influencing downstream neurons than more regular spiking (*Hoebeek et al., 2005*; *Luthman et al., 2011*). Therefore, we initially expected that a more irregular spike train in a floccular Purkinje cell would drive a smaller eye movement than a more regular spike train at the same mean rate. Contrary to this prediction, mean eye velocity, averaged across the 500 ms optogenetic stimulation interval, was indistinguishable during the regular and irregular stimulus trains, after controlling for mean spike rate. Within the 500 ms stimulation interval, the regular and irregular stimulus trains

elicited different eye movement trajectories on a moment-to-moment basis; however, these differences were well explained by the different trajectories of the instantaneous Purkinje cell population firing rate during the regular versus irregular trains (*Figure 10*). The mean spike rate in a given time interval—either a few milliseconds or the entire 500 ms train—could predict the mean eye velocity in that same time interval, with no additional effect of irregularity within that interval. Thus the irregularity of Purkinje cell spiking in the flocculus can affect the fine temporal structure of movement through a rapid rate code, without the need to invoke a separate temporal code.

## Potential for different coding strategies in different regions of the cerebellum

It is possible that the impact of spike irregularity in Purkinje cells could vary across different zones or microzones of the cerebellum. There is substantial variation in the gene expression, firing rates, and properties of synaptic plasticity in the Purkinje cells in different areas of the cerebellar cortex (*Suvrathan et al., 2016*; *Cerminara et al., 2015*; *Wadiche and Jahr, 2005*; *Kim et al., 2012*; *Witter and De Zeeuw, 2015*), and in the Purkinje cells' target neurons in different deep cerebellar and vestibular nuclei (*Chan-Palay, 1977*; *Sugihara and Shinoda, 2004*; *Sugihara, 2011*; *Sekirnjak and du Lac, 2006*; *Pugh and Raman, 2006*; *Bagnall et al., 2007*; *McElvain et al., 2010*; *Shin et al., 2011*; *Low et al., 2018*). Our biophysical modeling indicates that variations in model parameters within a biologically plausible range could determine whether spike irregularity either does or does not influence the transmission of information by the Purkinje cells. Although recording and stimulation results indicate that the cerebellar flocculus operates in a region of parameter space where the impact of spike irregularity is negligible, even during highly synchronous population activity, the model demonstrates that changing a single parameter, such as the ratio of excitation to inhibition, could create sensitivity to spike irregularity. Thus, other parts of the cerebellum may operate in regions of parameter space where spike irregularity has an impact beyond what can be accounted for by the rapid rate code. Some support for this possibility is provided by a recent report that regular, synchronous optogenetic stimulus trains delivered to Purkinje cells of the cerebellar lobulus simplex were less likely to perturb ongoing locomotion than continuous optogenetic stimulation, which elicits less regular, less synchronous spiking (*Sarnaik and Raman, 2018*). However, the interpretation of such results will require careful consideration of the effects on the Purkinje cell population spike rate in the behaviorally relevant, causal time window. Our finding that the population spike rate can influence behavior with high temporal precision demonstrates that a clear understanding of how the rate code influences behavior on a moment-to-moment basis is critical for evaluating the possibility that additional, temporal coding might also influence behavior. Such analyses are more challenging for complex behaviors such as locomotion, hence additional research is required to determine what, if any, role spike irregularity or other forms of temporal coding play in the cerebellar control of such behaviors.

## Implications for understanding the neural code

The ability of the brain to encode sensory information, process it, and generate motor responses depends on the neural code(s) used to transmit information. Thus, understanding how information is transmitted through the rate and timing of spikes is central to understanding neural computation, and can also guide the development of more effective interventions to treat brain disorders (*Birdno et al., 2008*; *Birdno et al., 2012*; *Summerson et al., 2015*; *McConnell et al., 2016*; *Karamintziou et al., 2016*). There has been extensive analysis of how temporal patterns of neural activity encode information in different brain areas (reviewed in *Singer, 1993*; *Borst and Theunissen, 1999*; *Wang et al., 2008*; *Ainsworth et al., 2012*; *Gire et al., 2013*; *Uchida et al., 2014*). However, in most mammalian circuits, it has been difficult to determine whether and how information encoded in the precise timing of spikes is read out, because the link between neural activity and behavior is typically too complex or remote. We leveraged the close link between floccular Purkinje cell activity and eye movements to analyze the contribution of spike rate and irregularity to the behavioral output, and found evidence only for rapid rate coding. Nevertheless, our modeling results indicate that biologically realistic variations in parameters could allow the precise temporal pattern of presynaptic activity to influence the downstream response. Thus, the extent to which information

encoded in spike timing gets transmitted to the downstream circuitry may vary across synapses, and might even vary dynamically at the same synapses depending on the behavioral state.

# Materials and methods

**Key resources table**

| Reagent type (species) or resource | Designation | Source or reference | Identifiers |
|---|---|---|---|
| Strain, strain background (Mus musculus) | ChR2(H134R)-EYFP | Jackson Laboratory | RRID: IMSR_JAX:024109 |
| Strain, strain background (Mus musculus) | (Pcp2-cre)3555 Jdhu/J | Jackson Laboratory | RRID: IMSR_JAX:010536 |
| Software, algorithm | NEURON biophysical model by *Luthman et al., 2011*. | ModelDB | 144523 |

## Electrophysiology in monkeys and mice during oculomotor behavior

### Recording procedure in monkeys and mice during oculomotor behavior

Extracellular recordings were made from Purkinje cells in the cerebellar flocculus and ventral paraflocculus of four adult male rhesus monkeys (data previously published in *Raymond and Lisberger, 1997*; *Raymond and Lisberger, 1998*; *Kimpo et al., 2014* and *Ke et al., 2009*) and the flocculus of 63 adult C57BL/6 mice (data previously published in *Katoh et al., 2015*) during eye movement responses to a range of different combinations of vestibular and visual stimuli. Eye movements were recorded with the eye coil method at 500 Hz. Vestibular stimuli were provided by using a turntable to passively rotate the head-fixed animal about an earth-vertical axis. For monkeys, visual stimuli were provided by horizontal motion of a small bright spot (target, T, subtending 0.5° of visual angle) and/or high-contrast background (BG, a 20°×30° grid of 1.5°×1.5° black and white squares) projected onto a tangent screen that was 114 cm in front of the eyes. For mice, the visual stimulus was provided by a striped hemispherical dome with vertical black and white stripes, each subtending 7.5° of visual angle, which surrounded the animal.

All four monkeys were tested with the following set of visual-vestibular stimuli: (1) sinusoidal motion of the visual target alone at 0.5 Hz, ±10°/s to ±31.4°/s to evoke *visual tracking* (smooth pursuit) eye movements; (2) sinusoidal motion of the head and visual target together at 0.5 Hz, ±10°/s to ±31.4°/s to evoke *VOR cancellation* (attenuation of the normal VOR eye movements). In addition, two of the monkeys (monkeys D and E) were tested with the following set of visual-vestibular stimuli: (3) steps of constant head velocity at 15°/s for 500 ms, with the visual target and background moving together with the head ('×0 step') or (4) opposite to the head ('×2 step') (*Raymond and Lisberger, 1998*; *Raymond and Lisberger, 1997*; *Kimpo et al., 2014*). The other two monkeys (monkeys C and L) were tested with the following visual-vestibular stimuli in addition to smooth pursuit and VOR cancellation: sinusoidal vestibular stimulation at 0.5 Hz, ±10°/s in combination with motion of the visual target and background, where the target and the background moved together with each other and with the vestibular stimulus (5, '×0T/×0BG') or opposite to the vestibular stimulus (6, '×2T/×2BG'); or where the target and the background moved opposite to each other with either the target (7, '×0T/×2BG') or the background (8, '×2T/×0BG') moving together with the head; or where the background was stationary but the target moved either together with the head (9, '×0T/×1BG') or opposite to the head (10, '×2T/×1BG') (*Ke et al., 2009*). Data from monkeys D and E are plotted in *Figure 1A,B*. Data from all four monkeys were plotted in *Figure 1C–E* (*top graphs*), where each color represents a single condition as described above. In total, the following number of $E_iH_i$ cells were recorded and plotted in *Figure 1C–E* for each condition listed above: (1) n = 120; (2) 120; (3) 26; (4) 27; (5) 50; (6) 38; (7) 47; (8) 47; (9) 49; (10) 38; spontaneous: 33. Data from smooth pursuit and VOR cancellation in all four monkeys were used in the residual analysis (*Figure 2*).

Mice were tested with (1) sinusoidal motion of the striped dome alone (1 Hz, ±10°/s) to evoke the optokinetic reflex ('visual tracking') and (2) sinusoidal motion of the striped dome together with the

animal (1 Hz, ±10°/s) to evoke VOR cancellation (n = 33 $E_iH_i$ cells). Spontaneous activity was recorded in a subset of these cells (n = 8).

## Trial-averaged analysis of neural activity in monkeys and mice during oculomotor behavior

The relationship between Purkinje cell simple spike activity and eye movements during oculomotor behavior was analyzed using MATLAB (The MathWorks, Inc). Instantaneous spike rate was computed by convolving each spike with a Gaussian kernel with 10 ms standard deviation. Instantaneous local irregularity was computed by first calculating the CV2 for each pair of ISIs according to *Equation 1*, where $CV2_i$ is calculated from the three spike times $t_{i-1}$, $t_i$, and $t_{i+1}$. For each pair of ISIs, the instantaneous local irregularity, CV2(t), was then set to the value of $CV2_i$ from time $t_i - (t_i - t_{i-1})/2$ to $t_i + (t_{i+1} - t_i)/2$.

Purkinje cells in the flocculus display a range of responses during horizontal eye movements. We focused on Purkinje cells with the most common directional preference in the flocculus across species: cells that increase their firing during both ipsiversive eye motion ($E_i$; peak firing within ±90° of peak ipsiversive eye velocity during visual tracking), and ipsiversive head motion ($H_i$; peak firing within ±90° of peak ipsiversive head velocity during VOR cancellation), and which therefore encode ipsiversive gaze velocity ($E_iH_i$ cells; *Miles et al., 1980*; *Lisberger, 1994*; *Raymond and Lisberger, 1997*; *Katoh et al., 2015*). Purkinje cells in the flocculus that exhibited other preferences for head or eye motion (increased firing for contraversive eye or head motion, or no sensitivity to one of the two; non-$E_iH_i$ cells) were tested for antiphasic modulation of spike rate and irregularity during oculomotor behaviors (*Figure 1—figure supplement 2*), however, all other analyses were performed only on the $E_iH_i$ Purkinje cells, for which the largest number of cells were recorded.

Trial-averaged neural and behavioral responses were calculated by averaging eye velocity, gaze velocity, spike rate, and instantaneous CV2 across trials for each cell, and then across cells (*Figure 1A*, *Figure 1—figure supplement 1*), where a 'trial' consisted of one cycle of a sinusoidal stimulus, or, for 'step' stimuli, a set of one ipsiversive and one contraversive stimulus. To examine the relationship between gaze velocity and spike rate, gaze velocity was divided into evenly spaced bins (bin width 2°/s), and the average spike rate was calculated in each bin by first averaging within a cell (*Figure 1B*, *top*), then across the population of $E_iH_i$ Purkinje cells (*Figure 1C*). Similarly, the relationship between gaze velocity and spike irregularity was analyzed by binning gaze velocity, then computing the average CV2 for each gaze velocity bin, first in each cell (*Figure 1B*, *bottom*), and then across the population (*Figure 1D*). The correlation coefficient was calculated on the binned data for each cell individually, and then averaged across cells. The hypothesis that each correlation coefficient was significantly different than zero was assessed using a one sample t-test. Finally, the relationship between spike irregularity and spike rate was summarized by calculating the instantaneous CV2 and spike rate at each time within a trial, averaged across trials for each cell, then averaged across cells, and finally comparing plotting spike rate versus CV2 for each time within the trial (*Figure 1E*).

## Simulated inhomogeneous Poisson process

To determine whether the observed inverse relationship between spike rate and irregularity could be generated by a inhomogeneous Poisson process modified by a refractory period (*Holt et al., 1996*), the average rate of Purkinje cell activity recorded in monkeys during smooth pursuit was calculated by averaging across all trials and all cells. An artificial spike train was then generated to match this average firing rate using an inhomogeneous Poisson process with an absolute refractory period. Artificial interspike intervals were drawn from an exponential distribution offset by the duration of the refractory period. The method of thinning was then used to match firing rates: artificial spikes were removed with probability inversely proportional to the firing rate (*Lewis and Shedler, 1979*). The artificial spike trains and real spike trains were then analyzed identically. To calculate mean firing rate, spikes were convolved with a 10 ms sigma Gaussian filter and averaged across trials. To calculate mean CV2, the instantaneous CV2 was calculated as above and averaged across trials. The instantaneous relationship between firing rate and CV2 was compared by binning all pairs of ISIs based on their mean firing rate, $Rate_i = 2/(ISI_i + ISI_{i+1})$, and calculating the mean CV2 for all ISI pairs in each firing rate bin according to *Equation 1* (*Holt et al., 1996*).

## Moment-to-moment analysis of neural activity in monkeys and mice during oculomotor behavior

The moment-to-moment fluctuations in neural activity and eye movements during oculomotor behavior were analyzed by first binning the raw spike rate, spike irregularity (CV2), and eye velocity in 50 ms time bins. This bin width was chosen to be short enough to capture fluctuations in eye velocity, and long enough to allow any measured fluctuations in Purkinje cell activity to impact eye velocity, given delays arising from the downstream neural circuitry and the oculomotor plant. Residual spike rate, CV2, and eye velocity were calculated by subtracting the corresponding cycle-averaged spike rate, CV2, and eye velocity from the binned response on each cycle of the stimulus. Residuals were calculated for all $E_iH_i$ cells during the visual tracking and VOR cancellation conditions, which were recorded for all cells in both monkeys and mice. Since there was negligible variation in the vestibular stimulus (head velocity) from trial-to-trial, the residual eye velocity was equivalent to residual gaze velocity.

The ability of residual spike rate and residual spike irregularity to predict residual eye velocity was assessed using a linear mixed effects model, which is well-suited for data with repeated measures (here, measurements within individual cells and across trials for a given visual-vestibular stimulus) with potential correlations between observations within each cell or stimulus condition (*Oberg and Mahoney, 2007*). The model contained residual rate, residual CV2, and the interaction term as fixed effects, and full random effects for each cell and stimulus condition:

$$E_{ijk} \sim \beta_o Rate_{ijk} + \beta_1 CV2_{ijk} + \beta_2 Rate_{ijk} CV2_{ijk} + b_{0jk} Rate_{ijk} + b_{1jk} CV2_{ijk} + b_{2jk} Rate_{ijk} CV2_{ijk} + \epsilon_{ijk}$$

where $E_{ijk}$ is the residual eye velocity in the $i$th time bin in Purkinje cell $j$ for stimulus condition $k$, $\beta_0$ represents the fixed effect of spike rate on eye velocity, $\beta_1$ is the coefficient for the fixed effect of CV2 on eye velocity, $\beta_2$ is the coefficient for the fixed effect of the interaction term, $b_{0jk}$, $b_{1jk}$, and $b_{2jk}$ are the random effect coefficients for rate, CV2, and their interaction, respectively. Intercept terms were not included because the calculation of residuals removes the intercept. The model was fit using the *fitlme* function in MATLAB using the default maximum likelihood method. The resulting coefficients and confidence intervals are reported in *Supplementary file 1*. Significance was determined by computing the log-likelihood ratio between the full model (above) and a reduced model with either the fixed effect of spike rate ($\beta_o$), the fixed effect of CV2 ($\beta_1$), or the fixed interaction effect ($\beta_2$) removed. An alternate full model without the interaction term was also assessed with no change in the conclusion. The outcomes of the log-likelihood test reported in the Results agreed with the outcome of tests for significance of the individual fixed effects parameters in the full model, as assessed by a Wald test comparing the coefficient's estimated value with its estimated standard error.

To visualize the relationship between oculomotor output and spike rate and irregularity, the residuals were graphically summarized in two ways (*Figure 2*). First, each individual time bin was classified into low or high CV2, according to whether the CV2 residual fell into the lowest or highest third of the distribution for that cell. Data within each group (low or high CV2 bins) were then further binned according to the spike rate residual, and average eye velocity was computed for each bin, first for all data points within each cell, and then averaged across cells. Second, a complementary visualization was created with spike rate rather than CV2 as the categorical variable: each time bin was classified into low or high spike rate, according to whether the rate residual fell into the lowest or highest third of the distribution for that cell. Data were then binned according to their CV2 residual and average eye velocity was computed for each bin, first for each cell, and then averaged across cells.

## Optogenetic stimulation in mice

### Surgical procedures

All experimental procedures were approved by the Administrative Panel on Laboratory Animal Care at Stanford University. Experiments were performed on male and female adult (≥8 weeks old) mice. All mice were housed on a reversed 12 hr light/12 hr dark cycle, and experiments were conducted during the dark cycle. ChR2 was selectively expressed in Purkinje cells by crossing Ai32 mice conditionally expressing ChR2(H134R)-EYFP (*Madisen et al., 2012*; Jackson Laboratory #024109) with

mice expressing Cre under the control of the L7 promotor ((Pcp2-cre)3555 Jdhu/J; Jackson Laboratory #010536; *Barski et al., 2000*; *Witter et al., 2016*).

A head post was secured to the skull with three screws and dental cement to allow head restraint. Guide cannulas (C312G, Plastics One) were implanted bilaterally to target the flocculus, as previously described (*Nguyen-Vu et al., 2013*). Eye movements were recorded using magnetic eye tracking (*Payne and Raymond, 2017*). Briefly, a 0.75 mm x 2 mm (diameter x height) cylindrical neodymium magnet was implanted beneath the conjunctiva of the left eye and secured with VetBond. An angular magnetic field sensor (HMC1512, Honeywell Inc) was attached via dental cement to the head implant and used to detect the angle of the magnet, and thus the angle of the eye as it rotated in its socket. After surgery, mice were individually housed and were allowed to recover for at least 5 d before behavioral experiments.

## Optogenetic stimulation of Purkinje cells

Optogenetic stimulus trains were delivered unilaterally to the cerebellar flocculus of awake, head-restrained mice in a dark room, and the resulting eye movements were measured. Optical stimulation was delivered via a 250 μm, 0.66 NA optical fiber (Prismatix). The shaft of the optical fiber was covered with black heat shrink tubing (Vention Medical) and the dental cement head implant was painted black to prevent light leakage. Blue light at 450 nm was provided by a high-powered LED light source (Prismatix, Israel), and the current was adjusted to produce either 1 mW or 10 mW at the tip of the optical fiber. The tip of the optical fiber was acutely inserted through the surgically implanted cannula and gradually advanced using a micromanipulator until eye movement responses to light pulses were first detected.

Optogenetic stimulus trains consisted of brief, 1 ms pulses of light delivered in either regular or irregular temporal patterns. Stimulus trains were 500 ms in duration, with 3 s between the start of one train and the start of the next. For each stimulus train, the CV of the interstimulus intervals was either 0 (perfectly regular), 0.5, or 1. Irregular stimulus trains were generated in MATLAB by using a Gamma distribution to generate sequences of interstimulus intervals. For each stimulus train, interstimulus intervals were drawn randomly from a Gamma distribution with parameters adjusted to generate the target stimulus rate and CV. Interstimulus intervals less than a refractory period of 3 ms were discarded. The resulting stimulus train was checked for three criteria: 1) the total number of stimuli in the irregular train was exactly the same as in the regular train of the same rate; 2) the mean stimulus rate, calculated by taking the reciprocal of the mean interstimulus interval, was within 5% of the target rate; and 3) the CV was within 5% of the target CV of 0, 0.5, or 1. If any of these conditions were violated, the stimulus train was discarded, and a new train was drawn, until 60 different irregular trains were obtained that satisfied the conditions for each stimulus frequency and CV. In addition to the 60 *non-repeated* irregular trains, for CV = 1 (20, 60, and 100 Hz), one irregular stimulus train was repeated to provide a measure of trial-to-trial variability, and to allow visualization of the average Purkinje cell (*Figure 3D,E*) and eye movement responses (*Figure 4A,B*) to irregular trains. Thus, each block of an experiment consisted of 18 different types of stimulus trains: regular stimulus trains (CV = 0) at 20, 40, 60, 80, and 100 Hz, irregular *non-repeated* trains (CV = 0.5) at 20, 40, 60, 80, and 100 Hz, irregular *non-repeated* trains (CV = 1) at 20, 40, 60, 80, and 100 Hz, and irregular *repeated* trains (CV = 1) at 20, 60, and 100 Hz. The order of the stimulus trains was randomized within each block. For the irregular trains, summary statistics (*Figure 3F,G*; *Figure 4C–E*) were calculated on the *non-repeated* trains, for which each trial was a unique temporal stimulus pattern. Mutual information (*Figure 5A*) was calculated from the regular trains and the *repeated* irregular trains (20 Hz, 60 Hz, and 100 Hz). The linear temporal filter model (*Figure 5B–E*) was trained on the *non-repeated* irregular trains, and tested on the *repeated* regular and irregular trains.

## In vivo recordings from Purkinje cells during optogenetic stimulation

Extracellular electrophysiological recordings were performed to characterize the simple spike responses of Purkinje cells to optogenetic stimulation. A tungsten microelectrode (Microprobes for Life Sciences, MD) was inserted directly alongside an optical fiber, and advanced past the tip of the optical fiber until a Purkinje cell layer was encountered. Signals were digitized in Spike2 at a sample rate of 50,000 Hz and stored offline. Spike sorting was conducted offline using Plexon Offline Sorter (Plexon Inc) to implement template matching in combination with principle component visualization.

Spike sorting using the MountainSort algorithm (*Chung et al., 2017*) yielded similar results (not shown). Any cell that exhibited signs of damage, such as bistable firing, inconsistent spike amplitudes, or a prolonged or abnormal spike waveform, or that was insufficiently isolated, as assessed from the first two principle components, were excluded. Only cells that were well isolated for at least 5 repeats of each stimulus condition were included in the analysis. Recordings were made from a total of 41 neurons, including 15 well-isolated Purkinje cells (13 cells recorded during 1 mW stimulation and 7 cells recorded during 10 mW stimulation, including 5 cells recorded for both light intensities), 12 putative cerebellar interneurons (8 cells during 1 mW stimulation and 4 cells during 10 mW stimulation), and 16 cells that were excluded for insufficient isolation, in a total of six mice.

## Measurement of eye movements responses to optogenetic stimulation

Eye movement responses to optogenetic stimulation of Purkinje cells were recorded in 13 mice, including five mice whose eye movements were recorded during the electrophysiological recordings. Eye movements were evoked by unilateral stimulation of the flocculus while the mouse was head-fixed in a dark room. Stimulation experiments were performed on the left and right flocculus on separate days, with at least two days between experiments. Each flocculus was first tested using a higher light intensity (10 mW). An experiment was excluded from the analysis if no evoked eye movements could be detected, or if the initial and steady state eye movement responses to optogenetic stimulus trains were in opposite directions (7/26 flocculi excluded for 10 mW stimulation). The remaining 19 flocculi were tested using a lower light intensity (1 mW), with the same exclusion criteria (3/19 flocculi excluded during 1 mW stimulation). For two mice with multiple repetitions of the experiment on the same flocculus (2 and 3 days, respectively), data were combined across days before averaging.

Eye movements were recorded using a previously published magnetic eye tracking method (*Payne and Raymond, 2017*). A small magnet was implanted beneath the conjunctiva of one eye as described above, and a skull-mounted magnetic field sensor (Honeywell International, Inc) was used to detect changes in the magnetic field as the magnetic moved with the eye. Eye movement-related signals from the magnetic sensor were recorded in Spike 2 (Cambridge Electronic Design) at a sample rate of 1000 Hz. The magnetic eye tracking system was calibrated by simultaneously recording eye position with a dual-angle video-oculography system during 1–2 min of sinusoidal vestibular stimulation in the light (1 Hz, ±20°/s). Eye position obtained from video-oculography and voltages from the two channels of the magnetic sensor were each differentiated and smoothed with a 100 Hz low pass filter and then fit with sine waves. The channel of the magnetic sensor whose differentiated voltage signal was best fit by a sine wave was selected. A scale factor for this channel was calculated by dividing the amplitude of the sine wave fit for the eye velocity signal recorded using video-oculography by the amplitude of the sine wave fit for the magnetic sensor. After the 1–2 min calibration session, the video-oculography system was removed from the mouse's field of view and experiments were conducted using magnetic eye tracking alone.

Eye position signals were digitally differentiated to yield eye velocity. Saccades and movement artifacts were detected using an automatic velocity threshold, and trials or sinusoidal stimulus cycles with saccades or movement artifacts were excluded from the analysis. For plotting example traces, eye position and velocity data were smoothed with a 5 ms moving average.

## Optogenetic stimulation: data analysis

Neural activity and eye movements evoked by optogenetic stimulation were analyzed in MATLAB to obtain the means across trials for each time point within a trial, or averaged across the entire 500 ms stimulus train. Eye velocity data were normalized to the maximum mean velocity for each experiment to allow comparison across experiments.

## Mutual information

Mutual information between the Purkinje cell spike rate and eye velocity was calculated for different temporal smoothing windows. For each Purkinje cell recorded during 1 mW optogenetic stimulation, spike counts were binned in 1 ms intervals, and averaged across trials to yield trial-averaged spike rates with 1 ms resolution. The population spike rate was computed by averaging spike rates across Purkinje cells. Mutual information between the population spike rate and eye velocity was calculated

separately for the eye velocity responses in each behavioral stimulation experiment, and then averaged across experiments. Mutual information was calculated separately on data from regular stimulus trains (CV = 0; 20, 60, and 100 Hz) and *repeated* irregular stimulus trains (CV = 1; 20, 60, and 100 Hz).

Because spike rate and eye velocity are continuous variables, rather than discrete variables, the mutual information between spike rate and eye velocity was calculated using a kernel estimate, which allows each individual observation to inform the probability distribution over the width of the kernel (*Moon et al., 1995*). The kernel estimate was computed using the MATLAB Central File Exchange function *kernelmi* with a default kernel width determined by the number of data points, N: $H = (N + 1)/(\sqrt{12} \cdot N^{1+0.25})$.

To account for the delay between Purkinje cell activity and eye velocity, a time delay was added to the Purkinje cell activity before calculating mutual information. The calculation of mutual information was repeated with time delays of 0–10 ms, in 1 ms increments, and the delay that maximized the average total mutual information, 6 ms (for the unsmoothed data), was used for all subsequent calculations of mutual information.

To estimate the temporal decoding window, spike rate was smoothed by taking a moving average over progressively longer time windows, from 1 ms (no smoothing) to 20 ms, before calculating mutual information between smoothed spike rate and unsmoothed eye velocity. Similar results were obtained if eye velocity was also smoothed at the same window length (not shown).

## Linear temporal filter fits

A linear temporal filter was used to predict eye velocity on rapid timescales from Purkinje cell spike rates. For each experiment, a linear temporal filter, F, was calculated according to the following:

$$\hat{\mathrm{E}}_i[t] = \sum_{m=0}^{M} \mathrm{F}[m]\mathrm{P}_i[t - m]$$

where $\hat{\mathrm{E}}_i[t]$ is the predicted eye velocity at time point *t* within stimulus train with pattern *i*, $\mathrm{P}_i[t]$ is the Purkinje cell spike rate averaged across the population of recorded cells for stimulus train *i*, and *m* is the index over the duration of the filter. For each experiment, the training set for fitting the linear filter consisted of the single-trial eye velocity responses to *non-repeated* irregular stimulus trains (CV = 1; 20, 40, 60, 80, 100 Hz), *excluding* the single pattern that was repeated. The filter was calculated using total least squares linear regression, since the uncertainty in both eye velocity and neural activity would bias ordinary least squares linear regression towards small filter weights. The test set consisted of the trial-averaged eye velocity responses to regular (CV = 0; 20, 60, 100 Hz) and irregular (CV = 1; 20, 60, 100 Hz) repeated stimulus trains. For each stimulus train in the test set, eye velocity was predicted by convolving the linear filter with the population mean spike rate using the equation above. The RMSE of the eye velocity prediction from the model was then compared for regular and irregular stimulus trains from a given experiment. The relationship between predicted and actual eye velocity was linear (correlation coefficient 0.777 ± 0.038, n = 16; see example in *Figure 5D*); therefore a nonlinearity was deemed unnecessary. Likewise, no regularization was necessary.

The linear filters were fit using unsmoothed spike histograms and behavior, both with 1 ms temporal resolution. To compare the model predictions, spike rate and eye velocity in the test set were smoothed with a sliding window of 5 ms, based on the estimated temporal decoding window derived from the mutual information calculations. This smoothing window was chosen to aid visualization and was not critical; similar results were obtained without any smoothing of the test data (1 ms time bins; RMSE 16.9°±1.8° regular, 16.5°±1.7° irregular, p=0.23, n = 16, paired t-test; compare with results in *Figure 5E* obtained using the 5 ms smoothing window).

## Biophysical model

Simulations were performed in the NEURON simulation environment (*Hines and Carnevale, 1997*) and analyses were conducted in MATLAB. A biophysically realistic model of a Purkinje cell target neuron in the deep cerebellar nuclei was adapted from *Luthman et al. (2011)*. Other than altering the rate and temporal pattern of input spikes, the only parameters we changed in the model are

those described in the text: the presence or absence of short-term depression, and the strength of excitatory and inhibitory conductances. The model target neuron received inhibitory synaptic input from 50 Purkinje cells (*Person and Raman, 2012a*) through a total of 450 inhibitory synapses (nine synapses per Purkinje cell) and excitatory synaptic input from 150 mossy fibers. The 50 Purkinje cells each fired spontaneously at a rate of 60 sp/s (to approximately match the mean spontaneous firing rate of 62 sp/s in the population of Purkinje cells recorded during optogenetic stimulation), with a CV of 0.5 for the ISIs between spontaneous spikes (to approximately match the irregularity of spontaneous firing recorded in vivo; *Figure 1E*). The mossy fibers each fired spontaneously at a mean rate of 20 sp/s with a CV of 1. Spontaneous activity on each trial was generated by drawing a sequence of ISIs from a Gamma distribution, removing spikes within a refractory period, and accepting only ISI sequences meeting the following criteria: (1) the total number of spikes was equal to the target value defined by 60 sp/s for Purkinje cells, or 20 sp/s for mossy fibers, (2) the mean spike rate, calculated by taking the reciprocal of the mean ISI, was within 10% of the target rate of 60 sp/s (Purkinje cells) or 20 sp/s (mossy fibers), and (3) the actual CV was within 10% of the target value of 0.5 (Purkinje cells) or 1 (mossy fiber). The target neuron contained eight Hodgkin-Huxley style ion channel conductances: a fast sodium current, a persistent sodium current, mixed fast Kv3 and slow Kv2 delayed rectifying current, a high-voltage-activated calcium current, a low-voltage-activated calcium current, a calcium-gated potassium (Sk) current, a hyperpolarization-activated cyclic-nucleotide gated (HCN) current, and a tonic non-specific cation current promoting spontaneous spiking.

To simulate optogenetic stimulation, 'stimulus-driven' spikes were added to the spontaneous spikes in each model Purkinje cell. Stimulus trains identical to those used in vivo triggered a spike in each model Purkinje cell at a variable latency described by a Gaussian distribution with mean 1.24 ms and standard deviation 1.36 ms (truncated at 0 ms) mimicking the latency jitter observed in vivo (*Figure 3B,C*). Optogenetic stimulus-driven spikes were not allowed to occur within an absolute refractory period of 1 ms after a spontaneous spike, in accordance with the short ISIs observed during optogenetic stimulation in vivo, whereas spontaneous spikes were restricted by an absolute refractory period of 3 ms after any other spike. For the small number of cases where a stimulus-driven spike was supposed to occur within the refractory period, that spike was skipped and spiking resumed with the next spontaneous or stimulus-driven spike scheduled to occur outside the refractory period. Since the optogenetic stimulation (0–100 Hz) was added on top of spontaneous activity (60 sp/s), the total spike rate of the model Purkinje cells ranged from 60 sp/s to 160 sp/s. A minimum of 10 trials were run for each rate, irregularity, and parameter set.

Model parameters describing the strength of excitation and inhibition were systematically varied: the excitatory conductance (gE) was scaled from 0.5 to 2 times the original values of 200 pS for AMPA and 172 pS for NMDA peak conductance (fast + slow), and the baseline inhibitory conductance (gI$_0$) was varied from 0.5 to 1.5 times the original value of 1.6 nS (*Steuber et al., 2011*). The models were run either with short-term synaptic depression at the synapses from Purkinje cells to the target neuron (*Luthman et al., 2011*; *Shin et al., 2007*), or with no short-term depression. The inhibitory conductance in the model without short-term depression was adjusted to equal the steady state conductance at the maximum spike rate in the corresponding model with short-term depression.

To test the effects of irregularity during asynchronous Purkinje cell spiking, asynchronous model Purkinje cell activity was generated by driving each individual Purkinje cell with the same stimulus train as during the corresponding synchronous optogenetic simulation, but with the stimulus-driven spike train for each cell independently shifted circularly by a random delay between 0 and 500 ms (*Figure 8*).

## Random walk model

The random walk simulations (*Figure 9*) were conducted using an extremely simple model neuron that perfectly integrates its inputs. On each time step, the membrane potential of the model neuron changes by an amount drawn from the random variable $\sim N(\mu, \sigma)$, which represents the mean ($\mu$) and standard deviation ($\sigma$) of the net contribution of all excitatory and inhibitory synaptic and intrinsic conductances (arbitrary units). The analytic approximations for the random walk model with $\mu \geq 0$ are taken from Salinas and Sejnowski (2000):

$$r_{out}^2 \Delta t^2 \left( (V_{thresh} + \sigma)^2 - V_{reset}^2 \right) - r_{out} \Delta t \left( 2\mu V_{reset} + \sigma^2 \right) - \mu^2 = 0$$

where $r_{out}$ is the output rate, $\Delta t$ is the time step, $V_{thresh}$ is the threshold 'voltage' (in arbitrary units), $V_{reset}$ is the reset voltage, and $\mu$ and $\sigma$ are given above. $V_{min}$, the minimum voltage permitted during the random walk, is not explicit in the equation above but is assumed to be 0, and can be effectively lowered by increasing both $V_{thresh}$ and $V_{reset}$ in parallel. Parameters were chosen to either match the previously published simulations (*Salinas and Sejnowski, 2000*; *Figure 9*); *black:* $\Delta t$ = 1 ms, $V_{thresh}$ = 40, $V_{reset}$ = 20, $V_{min}$ = 0, $\mu$ = 1.5, $\sigma$ varied) or changed to either increase the E/I ratio ($\mu$ = 2.5, *red*), decrease the E/I ratio ($\mu$ = 0, *blue*), or eliminate the reflecting floor ($V_{min}$ = –inf, *cyan*).

## Statistical analysis

Statistical tests were conducted using MATLAB and GraphPad Prism. All tests were two-tailed, and the significance level was set at $p < 0.05$. Data are summarized as mean ± SEM. The D'Agostino-Pearson omnibus test (Prism) was used to assess normality of the data for ANOVAs and t-tests. Data were normally distributed in the recordings of the Purkinje cell response to optogenetic stimulation (*Figure 3*), with the exception of one extreme outlier. The outlier was detected using both the modified z-score of *Iglewicz and Hoaglin (1993)* (z-score averaged across conditions at 100 Hz stimulation: 10.3, recommended threshold: 3.5) and Grubbs' extreme studentize deviate test (*Grubbs, 1969*; $t_{11}$ = 8.9, p=$10^{-6}$). Because parametric statistics such as means are highly sensitive to such outliers, all statistics were repeated both with and without the outlier, and the outlier was not included in the summary figures provided in the main text.

Two-way repeated measures ANOVA were conducted in Prism to assess the impact of optogenetic stimulus rate and irregularity (independent variables) on Purkinje cell spike rate and irregularity (*Figure 3G*). If the interaction term was significant, post-hoc comparisons were performed using the Tukey correction for multiple comparisons. A linear mixed effects model was conducted in MATLAB to assess the relationship between either Purkinje cell spike rate or stimulus rate and irregularity on eye velocity, since slightly different Purkinje cell spike rates were evoked by regular v. irregular stimulation at each frequency (*Figure 4*). The linear mixed effects model predicted eye velocity from fixed effects of Purkinje cell rate or stimulus rate, stimulus irregularity condition, and their interaction, with random intercepts by subject. Statistical significance was assessed using a likelihood ratio test with either stimulus irregularity condition removed or the interaction term removed. Paired t-tests were conducted in MATLAB to compare RMSE of the linear temporal filter predictions for the regular and irregular trains (*Figure 5*).

A linear mixed effects model was implemented in MATLAB for analyzing the correlation between Purkinje cell spike rate, CV2, and eye velocity, and statistical significance was assessed using a likelihood ratio test (see above). The residual data were not normally distributed, however, comparisons of means using linear models are robust to deviations of normality when the sample size is sufficiently large (*Lumley et al., 2002*; *Gelman and Hill, 2007*).

## Acknowledgements

We thank Mark Goldman, Jay Bhasin, John Huguenard, Sriram Jayabal, Jason Bant, Lane McIntosh, and Donald Payne for helpful discussions and comments on the manuscript; Soon-Lim Shin for early conceptual input; Akira Katoh and Michael Ke for sharing their published electrophysiology data; Max Gagnon for assistance with data analysis; Bob Schneeveis for construction of custom apparatus; and James Dang for technical assistance. This work was supported by the US National Institutes of Health (R01 DC04154 and R01 NS072406 JLR; P30 NS069375); the National Science Foundation Graduate Research Fellowship (DGE-114747; HLP); the National Science Foundation fellowship for Stanford Mind Brain Computation IGERT (0801700; HLP); the Stanford DARE (Diversifying Academia, Recruiting Excellence) Doctoral Fellowship Program (HLP); and the Academy of Finland (decision Nos. 315795 and 320072; TM).

# Additional information

## Competing interests
Jennifer L Raymond: Reviewing editor, *eLife*. The other authors declare that no competing interests exist.

## Funding

| Funder | Grant reference number | Author |
|---|---|---|
| National Science Foundation | DGE-114747 | Hannah L Payne |
| Stanford University | DARE Doctoral Fellowship Program | Hannah L Payne |
| National Science Foundation | 0801700 | Hannah L Payne |
| Academy of Finland | 315795 | Tiina Manninen |
| Academy of Finland | 320072 | Tiina Manninen |
| National Institutes of Health | R01-DC04154 | Jennifer L Raymond |
| National Institutes of Health | R01-NS072406 | Jennifer L Raymond |
| National Institutes of Health | P30-NS069375 | Jennifer L Raymond |

The funders had no role in study design, data collection and interpretation, or the decision to submit the work for publication.

## Author contributions
Hannah L Payne, Conceptualization, Data curation, Software, Formal analysis, Investigation, Visualization, Methodology, Writing—original draft, Writing—review and editing; Ranran L French, Software, Formal analysis; Christine C Guo, Conceptualization, Investigation; TD Barbara Nguyen-Vu, Conceptualization; Tiina Manninen, Conceptualization, Software, Formal analysis; Jennifer L Raymond, Conceptualization, Resources, Supervision, Funding acquisition, Investigation, Project administration, Writing—review and editing

## Author ORCIDs
Hannah L Payne https://orcid.org/0000-0003-4625-5706
Ranran L French http://orcid.org/0000-0002-7591-0223
Christine C Guo http://orcid.org/0000-0003-1530-0172
TD Barbara Nguyen-Vu http://orcid.org/0000-0002-4708-1982
Tiina Manninen https://orcid.org/0000-0002-0456-1185
Jennifer L Raymond http://orcid.org/0000-0002-8145-747X

## Ethics
Animal experimentation: All procedures complied with the recommendations in the Guide for the Care and Use of Laboratory Animals of the National Institutes of Health. Animals were handled strictly according to the guidelines of the institutional animal care and use committee (IACUC) at Stanford University, accredited by the Association for Assessment and Accreditation of Laboratory Animal Care International (AAALAC). Protocols were approved by the Administrative Panel on Laboratory Animal Care at Stanford University (protocol #9143). All surgery was performed under isoflurane anesthesia, and every effort was made to minimize animal suffering.

## Decision letter and Author response
Decision letter https://doi.org/10.7554/eLife.37102.034
Author response https://doi.org/10.7554/eLife.37102.035

## Additional files

### Supplementary files

• Source code 1. Source code for *Figure 1*. Requires *Figure 1—source data 1* and *Figure 1—source data 2*.
DOI: https://doi.org/10.7554/eLife.37102.023

• Source code 2. Source code for *Figure 2*. Requires *Figure 1—source data 1* and *Figure 1—source data 2*.
DOI: https://doi.org/10.7554/eLife.37102.024

• Source code 3. Source code for *Figure 3*. Requires *Figure 3—source data 1*.
DOI: https://doi.org/10.7554/eLife.37102.025

• Source code 4. Source code for *Figure 4*. Requires *Figure 4—source data 1*.
DOI: https://doi.org/10.7554/eLife.37102.026

• Source code 5. Source code for *Figure 5*. Requires *Figure 5—source data 1*.
DOI: https://doi.org/10.7554/eLife.37102.027

• Source code 6. Source code for *Figure 6*.
DOI: https://doi.org/10.7554/eLife.37102.028

• Source code 7. Source code for *Figures 7* and *8*.
DOI: https://doi.org/10.7554/eLife.37102.029

• Source code 8. Source code for *Figure 9*.
DOI: https://doi.org/10.7554/eLife.37102.030

• Supplementary file 1. Results from the linear mixed effects model used to predict residual eye velocity from residual Purkinje cell spike rate and irregularity. Fixed effect coefficients ($\beta_0$, spike rate; $\beta_1$, CV2; $\beta_2$, interaction between rate and CV2), 95% confidence intervals (CI), and p-value (F-test) for the fits to data from monkeys and mice are shown.
DOI: https://doi.org/10.7554/eLife.37102.031

• Transparent reporting form
DOI: https://doi.org/10.7554/eLife.37102.032

### Data availability

Supplementary files contain code and data to replicate the major components of all experimental figures, and source code has been provided for all model figures.

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
