## [Decision Letter]

Thank you for submitting your article "Cerebellar Purkinje cells control movement with a rapid rate code that is invariant to spike regularity" for consideration by *eLife*. Your article has been reviewed by three peer reviewers, and the evaluation has been overseen by Rich Ivry as the Senior Editor. The following individual involved in review of your submission has agreed to reveal his identity: David J Herzfeld (Reviewer #1).

The reviewers have discussed the reviews with one another and the Reviewing Editor has drafted this decision to help you prepare a revised submission.

Summary:

This manuscript addresses the question of how the regularity or irregularity of Purkinje cell firing contributes to motor behavior through a systematic analysis of the properties of floccular Purkinje cell firing rates during eye movements in primates and mice, including optogenetic (synchronous) stimulation of Purkinje cells with manipulations of regularity. The results provide evidence that regularity is not a primary variable in controlling oculomotor output.

Essential revisions:

All the reviewers were favorable about the caliber of the experiments and the extent of data gathered.

1) The most fundamental criticism that was agreed upon in an extensive discussion was that the dichotomizing of ideas like regularity/irregularity, rate/time, synchronous/asynchronous set up somewhat arbitrary oppositions that may not always be meaningful. For instance, it is not clear that it makes sense to refer to rate codes on the time scale of <2 spikes (please see reviewers' comments below), and consequently some of the interpretations and conclusions are overstated. To quote from the reviewers' discussion, "As written, it is possible to come away with the conclusion that all of this temporal coding business is refuted and we should just think about a rate code. The issue that the 'temporal code' can only be interrogated experimentally if we have access to coordinated patterns of PCs converging onto a [target] cell seems like an important one for the authors to address." More precise definitions and resulting interpretations would make the content more accurate as well as more accessible, and would help readers avoid conflating ideas (e.g., regularity / temporal coding / synchrony) that are distinct.

2) There was also a strong consensus that the efforts to overturn the pathological irregularity hypothesis were distracting from the main focus of the paper, since the results do not actually address pathologies and thus the link is somewhat weak. Although it is reasonable to cite the work on pathology as a motivation for the study, the reviewers generally agreed that emphasizing the positive outcomes of the study in the context of non-disease states would greatly improve the manuscript.

Another quote from the discussion expressed these points well and is excerpted and slightly paraphrased here for conciseness: "What the pathological papers show is that the average PC firing over many trials seems normal in animal models while the firing of single PCs during single trials is very irregular ('noisy'); this noise could potentially not average out over the whole population converging on a CN neuron and yield improper output. [This study] suggests that the time scale where the 'averaging out' should take place is in the sub-5ms range (at least in the absence of mf inputs). [With] the current state of literature (including this manuscript), it seems possible that the population instantaneous rate has to be coordinated/ specified/ learned at the 3-4ms time scale." The issues raised were that disproving the "pathological irregularity" hypothesis would require showing that the same eye velocity is obtained with regular and irregular spike trains. Figure 4 seems to show instead that while average results are similar, single trials indicate that temporal structure (rather than average rate) matters to the specific movements.

*Reviewer #1:*

- In the analyses surrounding Figure 2A-B (and Figure 2—figure supplement 1), please statistically state that the residuals of rate and CV2 are not correlated for both mice and monkeys. Given the results in Figure 2B, you should be able to perform a similar analysis to Figure 1B (significant correlation between the residual eye velocity and residual firing rate timeseries) without the need to separate the CV2 residuals into "low" and "high" bins. You could then statistically test the actual timeseries relationship by computing the coefficients of a multiple linear regression model with an output of eye velocity and show no significant effect of CV2. Such an analysis should have more power than the current segregation method.

- In Figure 5, it would be nice to see the kernel fit to 60Hz regular stimulation as well (overlayed on the current "irregularly fit" kernel). These two kernels should be visually the same.

- The linear filter in Figure 5B systematically underestimates the steady-state amplitude of the velocity response to regular stimulation (and to some extent, irregular stimulation) at high frequencies (Figure 5C, 100Hz). To me, this would seem to suggest that some low-pass filtering must be occurring downstream of the Purkinje cells. I would be interested to see the optimal linear filters fit to the 20Hz, 60Hz and 100Hz regular firing data. I imagine that the kernels would look different at the various frequencies – potentially suggesting that spike timing still matters, but only for relatively short ISIs (i.e., the net eye position after 250ms stimulation at 150Hz followed by 250ms stimulation at 50Hz may look different than position after a 500ms stimulation at 100Hz).

- I think that the authors should be very careful about wading too far into a debate on rate versus temporal codes. In particular, I find that the phrase "rapid rate code" goes slightly too far in the current manuscript. It is clear from Figure 4 and the temporal filter in Figure 5 that a single optical pulse (or, synonymously, spike) has a clear effect on motor output largely independent of spike timing. A single spike, in my mind, cannot have a rate – rate is defined either relative to the previous spike (ISI) or as the average within a time bin. I would advise the authors to focus the prose on "regularity" when discussing the contributions of single spikes rather than "rapid rates".

*Reviewer #2:*

In its state, the manuscript sounds over-confident in disproving a role of (poorly defined) increased irregularity of discharge in motor dysfunction. Negative results are tricky to establish, and a combination of a) study of natural irregularity (which may not recapitulate at all the pathological irregularity), b) massive optogenetic stimulations (providing very artificial over-synchronous patterns of firing), c) numerical models, falls a bit short of a final proof. While this manuscript brings a lot of very interesting experimental and modeling work, I feel it would be greatly improved by playing down the debate over regularity to the benefit of other interesting results (fluctuation of CV2 during a task in normal animals, tight temporal coupling of (synchronized) PC and eye movements, STP in the floccular output, etc.), rather than extrapolate to pathological conditions which are in fact barely studied in the study.

My main concerns are the following:

1) Irregularity is not really defined, and the debate not clearly phrased: Purkinje cells in normal brain in vivo are certainly not perfectly rhythmic, so we can already state that good cerebellar function is achieved with somewhat irregular Purkinje cell discharge (a position certainly agreed by most proponents of the pathological irregularity stand). Irregularity in the manuscript seems to refer only to high values of the (simple and imperfect) measures of the CV or the CV2 of the interspike intervals, which are respectively more relevant to stationary or non-stationary cases. These values are often reported for PC recordings (normal and pathological conditions) but they provide limited insight into the temporal structure of the spike trains and into the distribution of ISI: e.g. very high CV2 may be obtained for a cell producing very regular high-frequency doublets of spikes (signal perfectly periodic with only 2 possible ISIs); conversely, a very skewed distribution of ISI may be associated with very small CV2 if the successive ISIs are ordered (progressively accelerating/decelerating neuron). Whether the surrogate irregular patterns used in the study are "pathological" is not justified; they may indeed barely recapitulate the actual disrupted firing patterns observed in motor disorders and thus strongly limits the relevance of this study to test whether these patterns cause motor dysfunction. This is particularly critical for the modeling of asynchronous Purkinje cells (which is indeed the only part of the manuscript which could estimate the pathological contribution of irregular discharge). Finally, for evident reason of synaptic summation, synchronous and asynchronous irregularity in presynaptic neurons are going to affect very differently the target neurons; the manuscript is moving between single cell (optokinetic data)/synchronous (optogenetic and model)/asynchronous (model only) situations and barely acknowledges (they are identically named) their profound heterogeneity (nor discuss their differential relevance to pathological conditions). For example, sentence in the fourth paragraph of the subsection “No evidence for an influence of Purkinje cell spike regularity on motor control”,implicitly extends results from synchronous irregularity to any form of irregularity.

2) The first part of the manuscript studies whether the natural irregularity of PC is correlated with variability in movements. The main difficulty to interpret these data is for the negative result (residual CV2 not correlated with residual eye movements). Indeed, when the rate of a single Purkinje cell correlates with movement, this may either reflect the contribution of that single PC (in which case one can interpret the lack of contribution of fluctuation of the regularity of this cell), or that it reflects a global shift of excitation of the population of PC controlling the movement (and it may not matter that a single PC is a bit more irregular; I suspect that the model would confirm that). The Discussion acknowledges this issue but concludes from optogenetic data (synchronous PC stimulation) that this is not the case; however, optogenetically-driven synchrony deeply affects the impact of PC firing onto the cerebellar nuclei (Sarnaik and Raman, 2018). It is therefore not possible to conclude that the lack of correlation between residual CV2 and residual movements results is not due to the fact that these residual CV2 fluctuations are not correlated at the level of the population.

Related points:

a) The irregularity (as measured by the CV2) is naturally strongly anti-correlated with the average firing rate of neurons with a refractory period. This can be easily verified with Poisson trains trimmed from the short (<3ms) ISIs that the CV2 drops from ~0.9 for 20Hz train to ~0.5 for 150Hz cell. The manuscript states (subsection “Mean spike rate and regularity both vary with oculomotor behavior”) that the variability observed is not due to this simple effect but does not provide the demonstration (so the changes of CV2 may simply reflect the leftward shift of the ISI distribution shape -with a fixed refractory period- along the task, and the residual would be just stochastic fluctuations).

b) Please assess statistically the independence of the residuals of CV2 and residuals of rate (Figure 2—figure supplement 1).

c) Given the link between them, a mixed effect model with the interaction term between CV2 and rate would have been useful

3) The second part of the paper studies the coupling of PC firing to eye movement in a well-controlled, but very artificial configuration: no ongoing movement (and thus likely very little mossy fiber activity in the cerebellar nuclei), over-synchronous likely non physiological PC firing (very brief and efficient stimulations). Overall the results presented demonstrate very nicely that in these conditions, eye movements may follow very tightly arbitrary patterns of synchronous PC and they even provide accurate quantification on time scales of this drive. However, since the debate on PC regularity does not at all assumes synchronized PC firing patterns, this brings little in the debate. These data however provide a strong and original assessment of the PC-movement coupling capacities. The relevance to physiological conditions should be discussed, or better, explored in the context of a task (perturbation of movements), where more physiological mossy fiber activity would be present and using lighter stimulations.

4) The meaning of the I_irreg_ index is a bit difficult to connect with movement control. Since the link between target neuron and eye movement has been modeled, why not directly assess the error (or variance) of eye velocity as a function of irregularity?

5) The notion of "rate code" is a little unclear when it operates at the time scale of the code shorter than the ISIs (= the time to define an instantaneous firing rate)! This sounds more like a fast population code.

*Reviewer #3:*

No major comments for revision.

---

## [Author Response]

Essential revisions:All the reviewers were favorable about the caliber of the experiments and the extent of data gathered.1) The most fundamental criticism that was agreed upon in an extensive discussion was that the dichotomizing of ideas like regularity/irregularity, rate/time, synchronous/asynchronous set up somewhat arbitrary oppositions that may not always be meaningful. For instance, it is not clear that it makes sense to refer to rate codes on the time scale of <2 spikes (please see reviewers' comments below), and consequently some of the interpretations and conclusions are overstated. To quote from the reviewers' discussion, "As written, it is possible to come away with the conclusion that all of this temporal coding business is refuted and we should just think about a rate code. The issue that the 'temporal code' can only be interrogated experimentally if we have access to coordinated patterns of PCs converging onto a [target] cell seems like an important one for the authors to address." More precise definitions and resulting interpretations would make the content more accurate as well as more accessible, and would help readers avoid conflating ideas (e.g., regularity / temporal coding / synchrony) that are distinct.

We could not agree more with the reviewers that precise definition and use of terminology is critical, and have revised the text to clarify definitions with more extended discussion of potential points of confusion regarding terminology:

We now include an explicit acknowledgement that regularity/irregularity lie on a continuum (Introduction, third paragraph). In addition, we have taken care to soften the dichotomy between regular/irregular and synchronous/asynchronous by using phrases like “more irregular” and “less synchronous” throughout the manuscript.

We have expanded our discussion of the terms ‘rate coding’ and ‘temporal coding’. Throughout the manuscript (Introduction, Results, and Discussion), we indicate that we have adopted the definition of temporal coding used by Theunissen and Miller, 1995, with temporal coding used to refer to the precise temporal pattern of spikes encoding information beyond that carried by changes in spike rate over behaviorally relevant timescales. We also explicitly state (Introduction, second paragraph, subsection “No evidence for additional influence of spike irregularity”, first paragraph) that temporal coding could potentially take many forms, and we focused on just one form, the irregularity of the interspike intervals, since previous work in the cerebellar field has suggested that the irregularity of spiking in individual Purkinje cells might play a major role in their control of downstream neurons and behavior, in addition to, or instead of, the rate of spiking in the Purkinje cells (reviewed in De Zeeuw et al., 2011).

We agree with the reviewers that on very short time scales (e.g., < 5 ms), the distinction between rate coding and temporal coding is less intuitive, and we are grateful to the reviewers for highlighting the need for more extended discussion of this issue, which we now include in the subsection “Rapid rate coding”. At the level of individual neurons, it may not make sense to talk about rate coding on a time scale shorter than the interspike interval, since one needs at least two spikes to compute a rate. However, we think it is perfectly reasonable to consider the rate code in a *population* of neurons on such rapid time scales, since the population of ~50 Purkinje cells converging on a given target neuron will *collectively* emit more than two spikes, on average, over time scales shorter than the ISI in an individual Purkinje cell (e.g., 5 ms). In the revised manuscript (subsection “Rapid rate code for eye movements”), we clarify that in our stimulation experiments, we are always manipulating the rate and irregularity of spiking in a large population of Purkinje neurons. A key, take home message of the manuscript is that over both short and long time scales (a few milliseconds to 500 milliseconds) consideration of the population spike rate alone is sufficient to account for the eye velocity observed across our experimental conditions, with no need to hypothesize an additional, distinct effect of spike irregularity.

We have added a new, summary figure to provide a visual summary of these ideas (Figure 10).

We think our manuscript provides a useful contribution to the discussion in the scientific literature about the effects of Purkinje cell spike irregularity by offering an alternative, or at least more nuanced, perspective on this issue.

We agree that “all of this temporal coding business” is not refuted by our findings, and explicitly acknowledge (subsection “No evidence for additional influence of spike irregularity”, subsection “Potential for different coding strategies in different regions of the cerebellum”) that there could be 1) other forms of temporal coding in the Purkinje cells, and/or 2) conditions where the irregularity of Purkinje cell spiking could have an influence on downstream neurons above and beyond the effects on spike rate. Indeed, we report in the manuscript (subsection “Biophysical model of Purkinje cell synaptic transmission”, eighth paragraph) that our modeling work identified a key parameter, namely the relative strength of Purkinje cell inhibition versus excitatory input to the target neurons, that could determine whether Purkinje cell irregularity has an additional effect on the target neuron beyond the effects of spike rate. In the revised manuscript, we include additional, new simulations(Figure 9, subsection “Random walk model of the effect of input variance on target neuron spiking”) showing that even in an extremely simple, random-walk simulation, the moment-by-moment *variability* of the total synaptic input converging on a target neuron sometimes affects the mean rate of target neuron output, and sometimes does not, depending on the parameters of the model. We acknowledge (subsection “Potential for different coding strategies in different regions of the cerebellum”) that certain parts of the cerebellum could operate in a region of parameter space where Purkinje cell irregularity affects the mean rate of their target neurons. Further, we include an alternate mechanism by which the level of spike irregularity can affect motor readout without needing to invoke a temporal code, by altering the reliability of the population-level spike rate code (new Figure 6). Thus, our results do not refute previous work on spike irregularity of the Purkinje cells, but rather highlight the need for more extensive study of the conditions under which spike irregularity contributes or does not contribute to the cerebellar control of behavior.

That said, our results offer an important counterpoint to previous conclusions that spike irregularity is a key component of the code used by Purkinje cells to transmit signals to downstream neurons (Hoebeek et al., 2005; Luthman et al., 2011; De Zeeuw et al., 2011). In contrast to this view, we show that our results on the control of eye movements by Purkinje cells in the cerebellar flocculus can be fully accounted for by the population spike rate. We would note that initially, we expected to find an effect of Purkinje cell spike irregularity, based on the seminal work on this topic in the cerebellar literature (Hoebeek et al., 2005; Walter et al., 2006; Shin et al., 2007; Luthman et al., 2011; Sarnaik and Raman, 2018) and we designed our experiments and looked hard to find such effects. However, despite our initial expectation, we could find no evidence for an effect of irregularity per se, beyond what could be accounted for by the population spike rate. Hence we were persuaded by our data (changed our minds, based on the experimental evidence) that the irregularity of spiking in individual Purkinje cells is not a significant component of the code used by the cerebellar flocculus to control normal eye movements, and that a remarkably rapid rate code can account for our experimental results. We have modified the text to clarify this specific conclusion about the floccular control of eye movements.

Our study highlights the importance of future work to determine the extent to which spike rate and irregularity each contribute to the control of other behaviors by Purkinje cells in other regions of the cerebellum(subsection “Potential for different coding strategies in different regions of the cerebellum”).Just as our findings about rate coding for the control of eye movements by floccular Purkinje cells do not necessarily refute previous claims about the contribution of irregularity in other cerebellar regions and other behaviors, our results indicate that caution is likewise merited in generalizing previous findings about spike irregularity. Moreover, our results highlight the importance of adequately controlling for spike rate when considering the effects of spike irregularity in the interpretation of previous and future experiments. By offering an alternative scientific view, we hope that our study will prompt additional, in-depth study of this central issue about the neural code used by the cerebellum.

We also agree with the reviewers that the interrogation of temporal vs. rate coding requires consideration of three key parameters (Figure 10): 1) the timing of individual spikes within the spike trains of individual neurons; 2) the firing rate in an individual neuron or the population, averaged over a behaviorally relevant time window; and 3) the level of synchrony across the population of neurons that converge onto a given target cell. Since current methods do not allow precise experimental control of all three parameters simultaneously, we have combined recording, stimulation, and modeling approaches to interrogate the distinct contributions of firing rate, irregularity, and synchrony. The manuscript discusses the complementary strengths and limitations of our recording, stimulation, and modeling approaches in supporting our specific conclusion about the role of spiking rate vs. irregularity in the control of eye movements.

2) There was also a strong consensus that the efforts to overturn the pathological irregularity hypothesis were distracting from the main focus of the paper, since the results do not actually address pathologies and thus the link is somewhat weak. Although it is reasonable to cite the work on pathology as a motivation for the study, the reviewers generally agreed that emphasizing the positive outcomes of the study in the context of non-disease states would greatly improve the manuscript.

As pointed out by the reviewers, our main focus was to test the role of spike irregularity in normal motor control, rather than under pathological conditions of disease models. As suggested by the reviewers, the revised manuscript emphasizes the positive outcome of the study: that the normal control of eye movements can be accounted for by a rapid rate code (Introduction, last paragraph, subsection “Random walk model of the effect of input variance on target neuron spiking”, last paragraph, Discussion, first paragraph).

We also acknowledge that the range of irregularity we tested in our stimulation experiments was designed to match the range observed in normal mice during oculomotor behavior, as measured in our recording experiments, and that our most irregular stimulation condition produced a mean CV of the interspike interval (0.82) that was less irregular than typically observed under pathological conditions (where CVs of 0.77 – 2.25 have been reported; Hoebeek et al., 2005; Gao et al., 2012; Stahl and Thumser, 2014; Mark et al., 2015; subsection “Rapid rate code for eye movements”, last paragraph).

Another quote from the discussion expressed these points well and is excerpted and slightly paraphrased here for conciseness: "What the pathological papers show is that the average PC firing over many trials seems normal in animal models while the firing of single PCs during single trials is very irregular ('noisy'); this noise could potentially not average out over the whole population converging on a CN neuron and yield improper output. [This study] suggests that the time scale where the 'averaging out' should take place is in the sub-5ms range (at least in the absence of mf inputs). [With] the current state of literature (including this manuscript), it seems possible that the population instantaneous rate has to be coordinated/ specified/ learned at the 3-4ms time scale."

Although our focus is on normal cerebellar physiology, the reviewers’ comments motivated us to conduct *additional simulations* to more rigorously test the potential connection between our findings in normal mice and previous findings of abnormal spike irregularity of Purkinje cells under the pathological conditions of disease models. These simulations support the idea that abnormally high levels of Purkinje irregularity observed under pathological conditions could potentially disrupt the normal control of behavior by degrading the precision of the rate code on the rapid time scales (of 3-5 ms), which our analysis suggests is the time scale at which the rate code is read out downstream (Figure 6; subsection “Rapid rate code for eye movements”, last paragraph). In the original manuscript, we had included this as a speculative Discussion point. The more formal quantification of this idea improves the manuscript by 1) providing an alternative explanation for the disrupted motor coordination associated with abnormally irregular Purkinje cell spiking (which can be tested against the explanation that pathological irregularity has an effect independent of the degraded precision of the rate code); and 2) providing a stronger connection between the effects of irregularity under normal and pathological conditions. We greatly appreciate the reviewers’ comments, which motivated us to conduct the additional simulations.

The issues raised were that disproving the "pathological irregularity" hypothesis would require showing that the same eye velocity is obtained with regular and irregular spike trains. Figure 4 seems to show instead that while average results are similar, single trials indicate that temporal structure (rather than average rate) matters to the specific movements.

We have revised the text describing the stimulation results illustrated in Figure 4 and Figure 5, to prevent misinterpretation of those results. The central finding from those experiments is that the mean spike rate in a given time interval – either a few milliseconds or the entire 500 ms train – can predict the mean eye velocity in that same time interval, and we could detect no additional effect of irregularity within that interval.

Over short time scales of a few milliseconds, the “temporal structure”, or trajectory, of the eye velocity was different for optogenetic stimulus trains with different levels of stimulus irregularity, however, each trajectory could be well accounted for by the temporal structure of the instantaneous firing rate during the train. Therefore, it would be a misinterpretation of Figure 4 to conclude that the “*temporal structure (rather than average rate) matters to the specific movements*”, because the eye velocity in each period of a few milliseconds is well explained by the temporal structure of the average firing rate over those few milliseconds (Figure 5). We have revised the relevant part of the Results section to make this important and potentially confusing point as clear as possible, in a couple of different ways.

Over the longer time scale of 500 ms, we found that stimulus trains with different levels of irregularity evoked the same eye velocity averaged across the train duration when controlling for spike rate. This was a fairly surprising result, since previous work (Luthman et al., 2011) would have predicted different mean eye velocities.

We have created a new figure to visually summarize our experimental and modeling results on the effects of spike irregularity, synchrony, and rate, on short (3-5 ms) and long (500 ms) time scales (Figure 10). This new, summary figure makes two central points:

1) On short time scales of 3-5 ms, both synchrony and irregularity impact the moment-by-moment population spike rate. As illustrated in Figure 10, greater synchrony and greater irregularity can each produce more moment-to-moment variations in the population spike rate. Therefore, our finding of a very fast readout of the population rate (i.e., that the behavior follows the Purkinje cell population rate with a resolution of 3-5 ms) suggests that Purkinje cell synchrony and irregularity could each influence movement via an alteration of the instantaneous spike rate being transmitted by the population. In other words, rapid rate coding is a candidate mechanism for Purkinje cell synchrony or irregularity to influence the target neurons and behavior, with no need to hypothesize an additional effect of synchrony or irregularity, beyond their alteration of population spike rate. Thus, our analysis of rapid rate coding provides a potential unification of disparate phenomena.

2) Over longer time scales of tens or hundreds of milliseconds, the predicted impact of irregularity and synchrony on the *mean*target neuron activity (averaged over tens-hundreds of milliseconds) depends on biophysical parameters such as the excitation/inhibition ratio. For a given mean Purkinje cell rate, Purkinje cell irregularity and synchrony may or may not have an additional, independent influence on the mean level of target neuron activity. Moreover, these two regimes are both found within physiologically realistic variations in synaptic parameters (Figure 7, Figure 8, Figure 9).

Reviewer #1:In the analyses surrounding Figure 2A-B (and Figure 2—figure supplement 1), please statistically state that the residuals of rate and CV2 are not correlated for both mice and monkeys.

A quantification of the correlation between residual rate and residual CV2 was added (Figure 2—figure supplement 1). The correlation is non-zero but sufficiently small (correlation coefficients of -0.24 and -0.29 for monkeys and mice, corresponding to r^2^ of 0.06 and 0.08, respectively) to allow the linear mixed effects model to distinguish their relationship to eye velocity (Tabachnick and Fidell, 2013).

Given the results in Figure 2B, you should be able to perform a similar analysis to Figure 1B (significant correlation between the residual eye velocity and residual firing rate timeseries) without the need to separate the CV2 residuals into "low" and "high" bins. You could then statistically test the actual timeseries relationship by computing the coefficients of a multiple linear regression model with an output of eye velocity and show no significant effect of CV2. Such an analysis should have more power than the current segregation method.

The separation of residuals into discrete "low" and "high" CV2 or rate bins (Figure 2) was done purely for the purpose of providing an intuitive visualization of the results. The relationship between rate, CV2, and eye velocity residuals was statistically assessed using a linear mixed effects model (subsection “Spike rate, but not irregularity, predicts moment-to-moment variations in eye velocity”, first paragraph, Materials and methods subsection “Moment-to-moment analysis of neural activity in monkeys and mice during oculomotor behavior”). The linear mixed effect model has the advantage over multiple linear regression that it can account for correlated random effects attributable to individuals or behavioral conditions. Apparently, the statistical analysis was not given sufficient prominence in the original text, causing the reviewers to overlook it, hence we have revised the text (subsection “Spike rate, but not irregularity, predicts moment-to-moment variations in eye velocity”, first paragraph) and Figure 2 legend to better highlight the statistical analysis and to emphasize that it was done without binning the data into the “low” and “high” rate and CV2 bins used for the visualization.

- In Figure 5, it would be nice to see the kernel fit to 60Hz regular stimulation as well (overlayed on the current "irregularly fit" kernel). These two kernels should be visually the same.

We calculated a kernel fit to the regular stimulation data, and confirmed the reviewer’s expectation that the two kernels are visually similar (Figure 5—figure supplement 1). Note that the kernels were fit to the results for stimulation at all frequencies tested (20, 40, 60, 80, 100 Hz), not just 60 Hz.

- The linear filter in Figure 5B systematically underestimates the steady-state amplitude of the velocity response to regular stimulation (and to some extent, irregular stimulation) at high frequencies (Figure 5C, 100Hz). To me, this would seem to suggest that some low-pass filtering must be occurring downstream of the Purkinje cells. I would be interested to see the optimal linear filters fit to the 20Hz, 60Hz and 100Hz regular firing data. I imagine that the kernels would look different at the various frequencies – potentially suggesting that spike timing still matters, but only for relatively short ISIs (i.e., the net eye position after 250ms stimulation at 150Hz followed by 250ms stimulation at 50Hz may look different than position after a 500ms stimulation at 100Hz).

If this is an essential point, we would appreciate clarification. We do not understand what is to be gained by fitting filters to individual frequencies, and we are concerned about the potential for overfitting if only a single regular frequency is used for the fit, which was part of the rationale for using data from irregular stimulation to fit the model.

- I think that the authors should be very careful about wading too far into a debate on rate versus temporal codes. In particular, I find that the phrase "rapid rate code" goes slightly too far in the current manuscript. It is clear from Figure 4 and the temporal filter in Figure 5 that a single optical pulse (or, synonymously, spike) has a clear effect on motor output largely independent of spike timing. A single spike, in my mind, cannot have a rate – rate is defined either relative to the previous spike (ISI) or as the average within a time bin. I would advise the authors to focus the prose on "regularity" when discussing the contributions of single spikes rather than "rapid rates".

The concepts of rate vs. temporal coding have been defined more concretely in the revised text (subsection “No evidence for additional influence of spike irregularity”, first paragraph). In addition, in the second paragraph of the subsection “Rapid rate code for eye movements” and subsection “Rapid rate coding”, we clarify that our discussion of a “rapid rate code” refers to a *population* of Purkinje cells. A single optical pulse does not induce a single spike in a single Purkinje cell, but instead produces a brief, strong increase in the probability of spiking across the population of activated neurons. We estimated the population spike rate by binning spikes in short time bins across trials and across sequentially recorded neurons, and found that this estimated population firing rate could predict eye velocity on a time scale of a few milliseconds (Figure 5), without an additional influence of spike irregularity—we can think of no better way of expressing this than “rapid rate code”.

Reviewer #2:In its state, the manuscript sounds over-confident in disproving a role of (poorly defined) increased irregularity of discharge in motor dysfunction. Negative results are tricky to establish, and a combination of a) study of natural irregularity (which may not recapitulate at all the pathological irregularity), b) massive optogenetic stimulations (providing very artificial over-synchronous patterns of firing), c) numerical models, falls a bit short of a final proof. While this manuscript brings a lot of very interesting experimental and modeling work, I feel it would be greatly improved by playing down the debate over regularity to the benefit of other interesting results (fluctuation of CV2 during a task in normal animals, tight temporal coupling of (synchronized) PC and eye movements, STP in the floccular output, etc.), rather than extrapolate to pathological conditions which are in fact barely studied in the study.

We have revised the Introduction and Discussion to clarify that our results are most relevant to understanding the contribution (or lack thereof) of Purkinje cell spike irregularity to normal sensorimotor processing in the normal cerebellum, rather than motor dysfunction in pathological conditions. As described above, we think it is valuable to explicitly consider potential connections between findings made in different contexts, hence we have added additional simulations to begin to bridge our study of spike irregularity in wild type mice with previous observations in mouse models of ataxia.

The reality of systems neuroscience is that “final proof” is rarely if ever possible; the best one can do is to provide convergent evidence from multiple approaches with complementary strengths and limitations. We designed our recording, stimulation, and modeling approaches with this in mind, and have revised the text of the Discussion to more effectively highlight how the strengths of each approach help to address the limitations of the others (subsection “Potential for different coding strategies in different regions of the cerebellum”).

My main concerns are the following:1) Irregularity is not really defined, and the debate not clearly phrased: Purkinje cells in normal brain in vivo are certainly not perfectly rhythmic, so we can already state that good cerebellar function is achieved with somewhat irregular Purkinje cell discharge (a position certainly agreed by most proponents of the pathological irregularity stand).

In the revised manuscript, we make the explicit point that there is no perfect spike regularity, even in normal Purkinje cells, only varying levels of irregularity (Introduction, third paragraph). Also, we now describe Purkinje cell spiking as less/more irregular, rather than writing in false dichotomous terms. However, in describing the stimulus trains, which we controlled, the terms regular stimulation and irregular stimulation seem appropriate and well defined, hence we continue to use them in that context.

Irregularity in the manuscript seems to refer only to high values of the (simple and imperfect) measures of the CV or the CV2 of the interspike intervals, which are respectively more relevant to stationary or non-stationary cases. These values are often reported for PC recordings (normal and pathological conditions) but they provide limited insight into the temporal structure of the spike trains and into the distribution of ISI: e.g. very high CV2 may be obtained for a cell producing very regular high-frequency doublets of spikes (signal perfectly periodic with only 2 possible ISIs); conversely, a very skewed distribution of ISI may be associated with very small CV2 if the successive ISIs are ordered (progressively accelerating/decelerating neuron).

To clarify the nature of the Purkinje cell spike trains driven by the stimuli, we now show sample ISI distributions (Figure 3E) in addition to the summary statistics.

Whether the surrogate irregular patterns used in the study are "pathological" is not justified; they may indeed barely recapitulate the actual disrupted firing patterns observed in motor disorders and thus strongly limits the relevance of this study to test whether these patterns cause motor dysfunction. This is particularly critical for the modeling of asynchronous Purkinje cells (which is indeed the only part of the manuscript which could estimate the pathological contribution of irregular discharge).

We agree that our study was not about pathology, and now make it clear that we focused on a range of irregularity that is relevant for normal sensorimotor signal processing in wild-type animals. As we acknowledged in the original manuscript, this range is considerably more regular than reported for the *tottering* mice modeled by Luthman et al., 2011. In our revised manuscript, we provide a more extensive comparison of the range of irregularity reported in a variety of ataxic (pathological) mouse lines with the range of irregularity we studied (subsection “Rapid rate code for eye movements”, last paragraph and subsection “Rapid rate coding can endow sensitivity to spike irregularity and synchrony”).

Finally, for evident reason of synaptic summation, synchronous and asynchronous irregularity in presynaptic neurons are going to affect very differently the target neurons; the manuscript is moving between single cell (optokinetic data)/synchronous (optogenetic and model)/asynchronous (model only) situations and barely acknowledges (they are identically named) their profound heterogeneity (nor discuss their differential relevance to pathological conditions). For example, the sentence in the fourth paragraph of the subsection “No evidence for an influence of Purkinje cell spike regularity on motor control”, implicitly extends results from synchronous irregularity to any form of irregularity.

We agree with the reviewer that the single cell recordings, the highly synchronous optogenetic stimulation, and the synchronous and asynchronous models differ in important ways, and have revised the text to clarify these distinctions, and the complementary advantages/limitations of each approach (Discussion).

2) The first part of the manuscript studies whether the natural irregularity of PC is correlated with variability in movements. The main difficulty to interpret these data is for the negative result (residual CV2 not correlated with residual eye movements). Indeed, when the rate of a single Purkinje cell correlates with movement, this may either reflect the contribution of that single PC (in which case one can interpret the lack of contribution of fluctuation of the regularity of this cell), or that it reflects a global shift of excitation of the population of PC controlling the movement (and it may not matter that a single PC is a bit more irregular; I suspect that the model would confirm that). The Discussion acknowledges this issue, but concludes from optogenetic data (synchronous PC stimulation) that this is not the case; however, optogenetically-driven synchrony deeply affects the impact of PC firing onto the cerebellar nuclei (Sarnaik and Raman, 2018). It is therefore not possible to conclude that the lack of correlation between residual CV2 and residual movements results is not due to the fact that these residual CV2 fluctuations are not correlated at the level of the population.

The optimal experimental design for testing the effects of spike rate, irregularity, and synchrony would be to systematically and precisely manipulate each of these variables independently of the others. Unfortunately, there is currently no experimental technique for simultaneously controlling all three spiking parameters with sufficient precision. This challenge has affected all studies to date, including the study of Sarnaik and Raman (Sarnaik and Raman, 2018), which focused on experimental manipulation of synchrony, with less control of spike rate and irregularity. Given such experimental limitations, the computational modeling approach used in our manuscript and by other investigators (Luthman et al., 2011; Steuber et al., 2011) provides a useful complement to experimental recording and stimulation approaches, affording the opportunity to independently manipulate rate, irregularity and synchrony with any desired level of precision.

In the text, we have noted both the strengths and caveats of each approach. As the reviewer suggests, and as we acknowledge in the original and revised manuscripts (Discussion, second paragraph), it is possible that the residual CV2 may *not* be correlated across neurons at the level of the population, and that if irregularity did matter under these natural conditions, the effect of a single neuron may simply be too small to detect with the residual analysis. It is the preponderance of evidence from recording, stimulation and computational approaches that drives our conclusion, not any single result or observation.

Related points:a) The irregularity (as measured by the CV2) is naturally strongly anti-correlated with the average firing rate of neurons with a refractory period. This can be easily verified with Poisson trains trimmed from the short (<3ms) ISIs that the CV2 drops from ~0.9 for 20Hz train to ~0.5 for 150Hz cell. The manuscript states (subsection “Mean spike rate and regularity both vary with oculomotor behavior”) that the variability observed is not due to this simple effect but does not provide the demonstration (so the changes of CV2 may simply reflect the leftward shift of the ISI distribution shape -with a fixed refractory period- along the task, and the residual would be just stochastic fluctuations).

The main focus of our manuscript is the downstream effects of spike irregularity, rather than the related but separate issue of the upstream causes of the level of irregularity. Nevertheless, we have added an additional figure (Figure 1—figure supplement 3,subsection “Mean spike rate and irregularity both vary with oculomotor behavior”, last paragraph) to address this issue. This figure directly compares the relationship between firing rate and CV2 observed in the data with the prediction of an inhomogeneous Poisson process modulated by an absolute refractory period. This refractory Poisson model produces an inverse relationship between irregularity and rate, as the reviewer suggests, yet does not quantitatively match the data, providing support for the idea that the refractory period is a contributor, but not the sole contributor to the inverse correlation between firing rate and CV2.

b) Please assess statistically the independence of the residuals of CV2 and residuals of rate (Figure 2—figure supplement 1).

This analysis is included in the revised manuscript (Figure 2—figure supplement 1; see response to reviewer #1 above).

c) Given the link between them, a mixed effect model with the interaction term between CV2 and rate would have been useful

The linear mixed effect model for the data illustrated in Figure 2 was modified to include an interaction term between CV2 and rate. The interaction term did not significantly improve predictions (assessed using a log likelihood ratio between the full model with the interaction term and the model without the interaction term), and did not change the significance of the impact of rate or CV2. The results are now incorporated in the text (subsection “Spike rate, but not irregularity, predicts moment-to-moment variations in eye velocity”).

3) The second part of the paper studies the coupling of PC firing to eye movement in a well-controlled, but very artificial configuration: no ongoing movement (and thus likely very little mossy fiber activity in the cerebellar nuclei), over-synchronous likely non physiological PC firing (very brief and efficient stimulations). Overall the results presented demonstrate very nicely that in these conditions, eye movements may follow very tightly arbitrary patterns of synchronous PC and they even provide accurate quantification on time scales of this drive. However, since the debate on PC regularity does not at all assumes synchronized PC firing patterns, this brings little in the debate. These data however provide a strong and original assessment of the PC-movement coupling capacities. The relevance to physiological conditions should be discussed, or better, explored in the context of a task (perturbation of movements), where more physiological mossy fiber activity would be present and using lighter stimulations.

We thank the reviewer for recognizing our contribution to understanding Purkinje cell-movement coupling as “strong and original”. We agree that it is important to consider the relevance of the stimulation experiments to physiological conditions. In the relevant region of the cerebellar cortex (the flocculus) and target nucleus (the vestibular nuclei), the mossy fibers are not silent in the absence of movement, but rather are right in the middle of their physiological activity range. Mossy fiber inputs to the flocculus and vestibular nuclei have spontaneous firing rates of ~40-100 Hz (Lisberger and Fuchs, 1978; Sadeghi et al., 2007; Lasker et al., 2008), and increase their firing during vestibular stimuli or eye movements in one direction, and decrease their firing for eye movements in the other direction. Likewise, the floccular Purkinje cells’ target neurons in the vestibular nuclei have fairly high firing rates in the absence of eye movements (subsection “Biophysical model of Purkinje cell synaptic transmission”, sixth paragraph), and encode eye velocity in opposite directions with bidirectional changes in rate. Thus, the network state during our optogenetic stimulation experiments, which were done in the absence of eye movements, should be roughly in the center of its normal (“physiological”) working range.

Nevertheless, in the revised text, we acknowledge the limitations inherent in stimulation experiments (subsection “Biophysical model of Purkinje cell synaptic transmission”, last paragraph and Discussion, second paragraph). These limitations were an important motivation for the complementary and more “physiological” recording experiments (Figure 1, Figure 2) which analyze the natural neural activity during oculomotor behavior, during which both the levels of Purkinje cell synchrony and the activity of mossy fibers and other neurons is normal. Together, the convergent evidence from these complementary recording and stimulation approaches make the case that the irregularity of Purkinje cell firing has no detectable impact on eye movements beyond what can be accounted for by the spike rate.

4) The meaning of the I_irreg_ index is a bit difficult to connect with movement control. Since the link between target neuron and eye movement has been modeled, why not directly assess the error (or variance) of eye velocity as a function of irregularity?

We have revised the text to clarify that we used the I_irreg_ index to capture the overall impact of irregularity on the mean eye velocity across a range of stimulation frequencies (subsection “Biophysical model of Purkinje cell synaptic transmission”, seventh paragraph). In this analysis, we were testing for effects on the mean eye velocity averaged across the duration of the train, rather than the variance of eye velocity, because previous modeling work (Luthman et al., 2011) had predicted an effect of irregularity on the mean firing rate in the Purkinje cells’ target neurons.

In the revised manuscript, we also include a new analysis of how increased irregularity may affect the precision (variance) of motor output by increasing the variability of the population rate code (new Figure 6).

5) The notion of "rate code" is a little unclear when it operates at the time scale of the code shorter than the ISIs (= the time to define an instantaneous firing rate)! This sounds more like a fast population code.

As described above, we have revised the manuscript to clarify terminology, and specify that rapid rate coding on time scales faster than an ISI in a single neuron is considered with respect to the average spike rate in the population (subsection “Rapid rate code for eye movements”, first paragraph; subsection “Rapid rate coding”; subsection “No evidence for additional influence of spike irregularity”).